# EVENTFLOW: FORECASTING CONTINUOUS-TIME EVENT DATA WITH FLOW MATCHING

## ABSTRACT

Continuous-time event sequences, in which events occur at irregular intervals, are ubiquitous across a wide range of industrial and scientific domains. The contemporary modeling paradigm is to treat such data as realizations of a temporal point process, and in machine learning it is common to model temporal point processes in an autoregressive fashion using a neural network. While autoregressive models are successful in predicting the time of a single subsequent event, their performance can be unsatisfactory in forecasting longer horizons due to cascading errors. We propose `EventFlow`, a non-autoregressive generative model for temporal point processes. Our model builds on the flow matching framework in order to directly learn joint distributions over event times, side-stepping the autoregressive process. `EventFlow` is likelihood-free, easy to implement and sample from, and either matches or surpasses the performance of state-of-the-art models in both unconditional and conditional generation tasks on a set of standard benchmarks.

## 1 INTRODUCTION

Many stochastic processes, ranging from consumer behavior (Hernandez et al., 2017) to the occurrence of earthquakes (Ogata, 1998), are best understood as a sequence of discrete events which occur at random times. Any observed event sequence, consisting of one or more event times, may be viewed as a draw from a temporal point process (TPP) (Daley & Vere-Jones, 2003) which characterizes the distribution over such sequences. Given a collection of observed event sequences, faithfully modeling the underlying TPP is critical in both understanding and forecasting the phenomenon of interest.

While multiple different parametric TPP models have been proposed (Hawkes, 1971; Isham & Westcott, 1979), their limited flexibility limits their application when modeling complex real-world sequences. This has motivated the use of neural networks (Du et al., 2016; Mei & Eisner, 2017) in modeling TPPs. To date, most neural network based TPP models are autoregressive in nature (Shchur et al., 2020a; Zhang et al., 2020), where a model is trained to predict the next event time given an observed history of events. However, in many tasks, we are interested not only in the next event, but in the entire sequence of events which is to follow. While these models can achieve high likelihoods, their performance in many-step forecasting tasks can be unsatisfactory due to compounding errors arising from the autoregressive sampling procedure (Xue et al., 2022; Lüdke et al., 2023).

Moreover, existing models are typically trained via a maximum likelihood procedure (see Section 3) which involves computing the CDF implied by the learned model. When using a neural model, computing this CDF necessitates techniques such as Monte Carlo estimation to properly compute the loss (Mei & Eisner, 2017). In addition, sampling from intensity-based models (Du et al., 2016; Mei & Eisner, 2017; Yang et al., 2022) is nontrivial, requiring an expensive and difficult to implement approach based on the thinning algorithm (Lewis & Shedler, 1979; Ogata, 1981; Xue et al., 2024).

Motivated by these limitations, we propose `EventFlow`, a generative model which directly learns the joint event time distributions, thus allowing us to avoid autoregressive sampling altogether. Our proposed model extends the flow matching framework (Lipman et al., 2023; Albergo & Vanden-Eijnden, 2023; Liu et al., 2023) to the setting of TPPs, where we learn a continuous flow from a reference TPP to our data TPP. At an intuitive level, samples from our model are generated by drawing a collection of event times from a reference distribution and flowing these events along a learned vector field. The number of events is fixed throughout this process, decoupling the event counts and their times, so that the distribution over event counts can be learned or otherwise specified.

Figure 1: All illustration of forecasting with our `EventFlow` method. The horizontal axis indicates the flow time $s$, and the vertical axis indicates the support of the TPP $\mathcal{T} = [0, T]$. We first encode the observed history $\mathcal{H}$ into an embedding $e_{\mathcal{H}} = f_{\theta}(\mathcal{H})$. At $s = 0$, we independently draw $n$ events in the forecasting window $[T_0, T_0 + \Delta T]$ from a fixed reference distribution, constituting a sample $\gamma_0$ from a mixed-binomial TPP. Each event can be thought of as a particle, which is assigned a velocity by a neural network $v_{\theta}(\gamma_s, s, e_{\mathcal{H}})$. Each particle flows along its corresponding velocity field until reaching its terminal point at $s = 1$, whereby we obtain a forecasted sequence $\gamma_1$.

See Figure 1 for an illustration. As our primary contribution regards the modeling of the event times themselves, we focus on unmarked point processes in this work. More specifically,

- We propose `EventFlow`, a novel generative model for temporal point processes. Our model is suitable for both unconditional generation tasks (i.e., generating draws from the underlying data TPP) and conditional generation tasks (e.g., forecasting future events given a history), and is able to forecast multiple events simultaneously.

- Our model provides a new perspective on modeling TPPs and sidesteps common pitfalls in existing approaches. In particular, the key idea of `EventFlow` is to decompose the generative process into a learned event count distribution and a generative model for the joint distribution of event times. Our model is likelihood-free during training, non-autoregressive, easy to sample from, and straightforward to implement.

- On standard benchmark datasets, `EventFlow` obtains uniformly strong performance on a multi-step forecasting task, and matches or exceeds the performance of state-of-the-art models for unconditional generation.

## 2 RELATED WORK

**Temporal Point Processes**  The statistical modeling of temporal point processes (TPPs) is a classical subject with a long history (Daley & Vere-Jones, 2003; Hawkes, 1971; Isham & Westcott, 1979). The contemporary modeling paradigm, based on neural networks (Du et al., 2016), typically operates by learning a *history encoder* and an *event decoder*. The history encoder seeks to learn a fixed-dimensional vector representation of the history of a sequence up to some given time, and the decoder seeks to model a distribution over the subsequent event(s).

Numerous models have been proposed for both components. Popular choices for the history encoder include RNN-based models (Du et al., 2016; Shchur et al., 2020a; Mei et al., 2019) or attention-based models (Zhang et al., 2020; Zuo et al., 2020; Yang et al., 2022). While attention-based encoders can provide longer-range contexts, this benefit typically comes at the cost of additional memory overhead. Similarly, a wide range of forms for the event decoder have also been proposed. The most common approach is to parametrize a conditional intensity function via a neural network. For instance, several authors (Mei & Eisner, 2017; Zuo et al., 2020; Zhang et al., 2020) model the conditional intensity using a parametric form inspired by the Hawkes process (Hawkes, 1971), and Du et al. (2016) model the (log-)conditional intensity through an affine function of the history embedding. Similarly, Okawa et al. (2019) model the conditional intensity using a mixture of Gaussian kernels.

Most closely related to our work are approaches which use generative models as decoders. These models often do not assume a parametric form for the decoder, enhancing their flexibility. For instance, Xiao et al. (2017b) propose the use of W-GANs to generate new events. Similarly, Shchur et al. (2020a) learn the distribution over the next inter-arrival time via a normalizing flow. Lin et al. (2022) benchmark several choices of generative models, including diffusion, GANs, and VAEs. Despite the flexibility of these models, these approaches are all autoregressive in nature, making them ill-suited for multi-step forecasting tasks. In contrast, Lüdke et al. (2023) propose a diffusion-style model which is able to avoid autoregressive sampling via an iterative refinement procedure.

Our work can be viewed as a novel approach for building flexible decoders for TPPs, extending flow matching to the setting of continuous-time event sequences. In contrast to prior work using generative models, our model is likelihood-free and non-autoregressive, achieving strong performance on long-term forecasting tasks. The work of Lüdke et al. (2023) is perhaps most closely related to ours, but we emphasize that the method of Lüdke et al. (2023) requires an involved training and sampling procedure. In contrast, our method is straightforward to both implement and sample from, while simultaneously outperforming existing approaches.

**Flow Matching** The recently introduced flow matching framework (or stochastic interpolants) (Lipman et al., 2023; Albergo & Vanden-Eijnden, 2023; Liu et al., 2023) describes a class of generative models which are closely related to both normalizing flows (Papamakarios et al., 2021) and diffusion models (Ho et al., 2020; Song et al., 2021). Intuitively, these models learn a path of probability distributions which interpolates between a fixed reference distribution and the data distribution. These models are a popular alternative to diffusion, providing greater flexibility in model design, with recent applications in image generation (Ma et al., 2024; Dao et al., 2023), DNA and protein design (Stark et al., 2024; Campbell et al., 2024), and point cloud generation (Buhmann et al., 2023; Wu et al., 2023). To the best of our knowledge, our work is the first to explore flow matching for TPPs.

## 3 AUTOREGRESSIVE TPP MODELS

We first provide a brief review of autoregressive point process models and discuss their shortcomings. Informally, one may think of an event sequence as a set $\{t_k\}_{k=1}^n$ of increasing event times. We will use $\mathcal{H}_t$ to represent the history of a sample up to (and including) time $t$, i.e., $\mathcal{H}_t = \{t_k : t_k \leq t\}$. Similarly, we use $\mathcal{H}_{t^-} = \{t_k : t_k < t\}$ to represent the history of a sample prior to time $t$. In the autoregressive setting, the time of a single future event $t$ is modeled conditioned on the observed history of a sequence. This is typically achieved by either directly modeling a distribution over $t$ (Shchur et al., 2020a), or equivalently by modeling a conditional intensity function (Du et al., 2016).

In the first approach, a conditional probability density of the form $p(t \mid \mathcal{H}_{t_n})$ is learned, allowing us to specify a joint distribution over event times $p(t_1, \ldots, t_n)$ autoregressively via $p(t_1, \ldots, t_n) = p(t_1) \prod_{k=2}^n p(t_k \mid \mathcal{H}_{t_{k-1}})$. Alternatively, we may define the *conditional intensity* $\lambda^*(t) := \lambda(t \mid \mathcal{H}_{t^-}) = p(t \mid \mathcal{H}_{t_n})/(1 - F(t \mid \mathcal{H}_{t_n}))$, where $F(t \mid \mathcal{H}_{t_n}) = \int_{t_n}^t p(s \mid \mathcal{H}_{t_n}) \, ds$ is the CDF associated with $p(t \mid \mathcal{H}_{t_n})$. Informally, the conditional intensity $\lambda^*(t)$ can be thought of (Rasmussen, 2011) as the instantaneous rate of occurrence of events at time $t$ given the previous $n$ events and given that no events have occurred since $t_n$. By integrating $\lambda^*(t)$, one can show that

$$F(t \mid \mathcal{H}_{t_n}) = 1 - \exp\left(-\int_{t_n}^t \lambda^*(s) \, ds\right) \qquad p(t \mid \mathcal{H}_{t_n}) = \lambda^*(t) \exp\left(-\int_{t_n}^t \lambda^*(s) \, ds\right) \quad (1)$$

and thus one may recover the conditional distribution from the conditional intensity under mild additional assumptions (Rasmussen, 2011, Prop 2.2).

**The Likelihood Function** Suppose we observe an event sequence $\{t_k\}_{k=1}^n$ on the interval $[0, T]$. The *likelihood* of this sequence can be seen loosely as the probability of seeing precisely $n$ events at these times. The likelihood may be expressed in terms of either the density or intensity via

$$L(\{t_k\}) = p(t_1, \ldots, t_n)\left(1 - F(T \mid \mathcal{H}_{t_n})\right) = \left(\prod_{k=1}^n \lambda^*(t_k)\right) \exp\left(-\Lambda^*(T)\right) \quad (2)$$

where the CDF term is included to indicate that no events beyond $t_n$ have occurred and $\Lambda^*(T) = \int_0^T \lambda^*(s)\,\mathrm{d}s$ is the total intensity. Autoregressive models are typically trained by maximizing this likelihood (Du et al., 2016; Mei & Eisner, 2017; Shchur et al., 2020a). We emphasize that this likelihood is not simply the joint event-time density $p_n(t_1, \ldots, t_n)$, as the likelihood measures the fact that no events occur after $t_n$.

It is worth noting that evaluating $L(\{t_k\})$ can be non-trivial in practice. For models which parametrize $\lambda^*(t)$ via a neural network (Du et al., 2016; Mei & Eisner, 2017), computing the total intensity $\Lambda^*(T)$ is often done via a Monte Carlo integral, requiring several forward passes of the model to evaluate $\lambda^*(t)$ at different values of $t$. Models which directly parametrize the density $p(t \mid \mathcal{H}_t)$ suffer from the same drawback when computing the corresponding CDF in Equation (2). Moreover, some approaches, such as the diffusion-based approach of Lin et al. (2022), are only trained to maximize an ELBO of $p(t \mid t_1, \ldots, t_n)$, and are thus unable to compute the proper likelihood in Equation (2).

**Sampling Autoregressive Models** In many tasks, we are interested not only in an accurate model of the intensity (or distribution), but also sampling new event sequences from the corresponding distribution. For instance, when forecasting an event sequence, we may want to generate several forecasts in order to provide uncertainty quantification over these predictions. However, sampling from existing autoregressive models can be difficult.

For instance, the flow-based model of Shchur et al. (2020a) requires a numerical approximation to the inverse of the model to perform sampling. Similarly, the diffusion-based approach of Lin et al. (2022) can require several hundred forward passes of the model to generate a single event time, rendering it costly when generating long sequences. Moreover, the predictive performance of autoregressive models is often unsatisfactory on multi-step generation tasks due to the accumulation of errors over many steps (Lin et al., 2021; Lüdke et al., 2023). This difficulty is particularly pronounced for intensity-based models (Du et al., 2016; Mei & Eisner, 2017; Zhang et al., 2020), where naively computing the implied distribution in Equation (1) is prohibitively expensive. Instead, sampling from intensity-based models is typically achieved via the thinning algorithm (Ogata, 1981; Lewis & Shedler, 1979). However, this algorithm has several hyperparameters to tune, is challenging to parallelize, and can be difficult for practitioners to implement (Xue et al., 2024).

## 4 EVENTFLOW

Motivated by the limitations of autoregressive models, we propose `EventFlow`, which has a number of distinct advantages over existing approaches. First, `EventFlow` directly models the joint distribution over event times, thereby avoiding autoregression entirely. Second, our model is likelihood-free, avoiding the Monte Carlo estimates needed to estimate the likelihood in Equation (2) during training. Third, sampling from our model amounts to solving an ordinary differential equation. This is straightforward to implement and parallelize, allowing us to avoid the difficulties of thinning-based approaches used in existing models. We build upon the flow matching (or stochastic interpolant) framework (Lipman et al., 2023; Albergo & Vanden-Eijnden, 2023; Liu et al., 2023) to develop our model. We begin below by focusing on the unconditional setting, and later discuss how to extend our method for conditional generation as necessary.

### 4.1 PRELIMINARIES

We first introduce some necessary background and notation. Let $\mathcal{T} = [0, T]$ be a finite-length interval. The set $\Gamma$ denotes the *configuration space* of $\mathcal{T}$ (Albeverio et al., 1998), i.e., the set of all finite counting measures on the set $[0, T]$. A point $\gamma \in \Gamma$ corresponds to a measure of the form $\gamma = \sum_{k=1}^n \delta[t^k]$, i.e., a finite collection of Dirac deltas located at event times $t^k \in \mathcal{T}$. A *temporal point process (TPP)* on $\mathcal{T}$ is a probability distribution $\mu$ over the configuration space $\Gamma$. Informally, $\mu$ represents a distribution over sequences $\gamma$ living in the configuration space $\Gamma$ which constitutes the set of valid sequences. We use $N : \Gamma \to \mathbb{Z}_{\geq 0}$ to denote the counting functional, i.e., $N(\gamma)$ is the number of events in the TPP realization $\gamma$.[1] While it is common to represent TPPs as distributions

---

[1]This can be thoughts of in terms of the counting process, i.e., $N(\gamma)$ corresponds to the value of the associated counting process at the ending time $T$, or the total number of events in $\gamma$ that have occurred in the interval $[0, T]$.

over random sets of event times, in our approach it will be more convenient to represent TPPs as random measures (Kallenberg et al., 2017).

We assume all TPP distributions are atomless (Kallenberg et al., 2017, Ch. 1), i.e., the probability of observing an event at any singleton is zero. In addition, we assume all TPPs are simple (Kallenberg et al., 2017, Ch. 2), i.e., no more than one event can occur simultaneously. Under these assumptions, a TPP $\mu$ can be fully characterized (Daley & Vere-Jones, 2003, Prop. 5.3.II) by a probability distribution which specifies the number of events and a *collection* of joint densities corresponding to the event times themselves. In a slight abuse of notation, we will write $\mu(n)$ for the corresponding distribution over event counts, and $\{\mu^n(t^1, \ldots, t^n)\}_{n=1}^\infty$ for the collection of joint distributions. In other words, for any given $n \in \mathbb{Z}_{\geq 0}$, the probability of observing $n$ events in the interval $\mathcal{T}$ is $\mu(n)$, and $\mu^n(t^1, \ldots, t^n)$ describes the corresponding joint distribution of event times. We further restrict each $\mu^n$ to be supported only on the ordered sets, so that we may assume $t^1 < t^2 < \cdots < t^n$.

Let $\mu_1$ represent the data distribution and $\mu_0$ represent a reference distribution. That is, both $\mu_0, \mu_1 \in \mathbb{P}(\Gamma)$ are TPP distributions. To construct our model, we will define a path of TPPs $\eta_s \in \mathbb{P}(\Gamma)$ which approximately interpolates from our reference distribution to our data distribution. Throughout, we use $s \in [0, 1]$ to denote a flow time and $t \in [0, T]$ to denote an event time. These two time axes are in a sense orthogonal to one another (see Figure 1).

## 4.2 BALANCED COUPLINGS

Our first step is to define a useful notion of couplings (Villani et al., 2009), allowing us to pair event sequences drawn from $\mu_0$ with those drawn from $\mu_1$. A *coupling* between two TPPs $\mu, \nu \in \mathbb{P}(\Gamma)$ is a joint probability measure $\rho \in \mathbb{P}(\Gamma \times \Gamma)$ over pairs of event sequences $(\gamma_0, \gamma_1)$ such that the marginal distributions of $\rho$ are $\mu$ and $\nu$. We say that the coupling $\rho$ is *balanced* if draws $(\gamma_0, \gamma_1) \sim \rho$ are such that $N(\gamma_0) = N(\gamma_1)$ almost surely. In other words, balanced couplings only pair event sequences with equal numbers of events. While we will later see how to interpolate between any two given event sequences, this coupling will allow us to decide *which* sequences to interpolate. In particular, a balanced coupling will allow us to pair sequences such that they always have the same number of events, allowing us to avoid the addition or deletion of events during both training and sampling and thus simplifying our model. We will use $\Pi_b(\mu, \nu)$ to denote the set of balanced couplings of $\mu, \nu$, and the following proposition shows $\Pi_b(\mu, \nu)$ is nonempty if and only if the event count distributions of $\mu$ and $\nu$ are equal, placing a structural constraint on the suitable choices of a reference measure.

**Proposition 1** (Existence of Balanced Couplings).
*Let $\mu, \nu \in \mathbb{P}(\Gamma)$ be two TPPs. The set of balanced couplings $\Pi_b(\mu, \nu)$ is nonempty if and only if $\mu(n) = \nu(n)$ have the same distribution over event counts.*

We provide a proof in Appendix B. In practice, we follow a simple strategy for choosing both the reference TPP $\mu_0$ and the coupling $\rho$. Suppose $q$ is any given density on our state space $\mathcal{T}$, e.g., a uniform distribution. We take $\mu_0$ to be a mixed binomial process (Kallenberg et al., 2017, Ch. 3) whose event count distribution is given by that of the data $\mu_1(n)$ and joint event distributions given by independent products of $q$ (up to sorting). That is, to sample from $\mu_0$, one can simply sample $n \sim \mu_1(n)$ from the empirical event count distribution followed by sampling and sorting $n$ i.i.d. points $t^k \sim q$. To draw a sample from our coupling $\rho$, we first sample a data sequence $\gamma_1 \sim \mu_1$, followed by sampling $N(\gamma_1)$ events independently from $q$ and sorting to produce a draw $\gamma_0 \sim \mu_0$. We call this coupling the *independent balanced coupling* of $\mu$ and $\nu$.

## 4.3 INTERPOLANT CONSTRUCTION

We now proceed to construct our interpolant $\eta_s \in \mathbb{P}(\Gamma)$. We will construct this path of TPPs via a local procedure which we then marginalize over a given balanced coupling. Here, we adapt standard flow matching techniques (Lipman et al., 2023; Tong et al., 2024) to design our sequence-level interpolants, but we emphasize that this is only possible under a balanced coupling as the number of events is fixed. To that end, let $\rho$ be any balanced coupling of the reference measure $\mu_0$ and the data measure $\mu_1$, and suppose $z := (\gamma_0, \gamma_1) \sim \rho$ is a draw from this coupling. As $\rho$ is balanced, we have $\gamma_0 = \sum_{k=1}^n \delta[t_0^k]$ and $\gamma_1 = \sum_{k=1}^n \delta[t_1^k]$ are both a collection of $n$ events. As TPPs are fully characterized by their joint distributions over event times, we will henceforth describe our procedure

for a fixed (but arbitrary) number of events $n$. First, we the define measure $\gamma_s^z \in \Gamma$ via

$$\gamma_s^z = \sum_{k=1}^n \delta[t_s^k] \qquad t_s^k = (1-s)t_0^k + st_1^k \qquad 0 \leq s \leq 1 \tag{3}$$

where we use the superscript $z$ to denote the dependence on the pair $z = (\gamma_0, \gamma_1)$. In other words, $\gamma_s^z$ linearly interpolates each corresponding event in $\gamma_0$ and $\gamma_1$. This defines a path $(\gamma_s^z)_{s=0}^1$ in the configuration space $\Gamma$ which evolves the reference sample $\gamma_0$ into the data sample $\gamma_1$.

In order to perform the marginalization step, we now lift this deterministic path $(\gamma_s^z)_{s=0}^1 \subset \Gamma$ to a path of TPP distributions $(\eta_s^z)_{s=0}^1 \subset \mathbb{P}(\Gamma)$. We define the point process distribution $\eta_s^z \in \mathbb{P}(\Gamma)$ implicitly by adding independent Gaussian noise to each of the events in $\gamma_s^z$. That is, a draw $\hat{\gamma}_s^z \sim \eta_s^z$ may be simulated via

$$\hat{\gamma}_s^z = \sum_{k=1}^n \delta\left[t_s^k + \epsilon^k\right] \qquad \epsilon^k \sim \mathcal{N}(0, \sigma^2). \tag{4}$$

In principle this means that the support of $\eta_s^z$ is larger than $\mathcal{T}$, but in practice we choose $\sigma^2$ sufficiently small such that this is not a concern. The addition of noise $\epsilon_k$ is instrumental in obtaining a well-specified model, but in practice the noise variance $\sigma^2$ is not a critical hyperparameter. We note that this noising step is typical in flow matching models (Lipman et al., 2023; Tong et al., 2024).

Finally, for any $s \in [0, 1]$, we define the marginal TPP measure $\eta_s$ via $\eta_s = \int \eta_s^z \, \mathrm{d}\rho(z)$. Observe that, by construction, the event count distribution $\eta_s(n)$ is given by $\mu_1(n)$ for all $s \in [0, 1]$. This path of TPP distributions $\eta_s$ approximately interpolates from the reference TPP $\mu_0$ at $s = 0$ to the data TPP $\mu_1$ at $s = 1$, in the sense that at the endpoints, the joint event time distributions $\eta_0^n(t^1, \ldots, t^n)$ and $\eta_1^n(t^1, \ldots, t^n)$ are given by a convolution of $\mu_0^n(t^1, \ldots, t^n)$ and $\mu_1^n(t^1, \ldots, t^n)$ with the Gaussian $\mathcal{N}(0, \sigma^2 I_n)$. As $\sigma^2 \downarrow 0$, it is clear that we recover a genuine interpolant (Tong et al., 2024).

We now shift our attention to the transport of a single event $t_s^k$ for a fixed $k$. Through the addition of Gaussian noise, we have constructed a path of Gaussian distributions $\mathcal{N}(t_s^k, \sigma^2)$ whose mean is determined by the location of the $k$th event at the flow time $s$. This transport of a Gaussian can be achieved infinitesimally through the constant vector field $v_s^k : [0, T] \to \mathbb{R}$ given by $v_s^k(t) = t_1^k - t_0^k$ (Tong et al., 2024). Thus, the evolution in (4) is generated by the vector field $v_s^z : \mathcal{T}^n \to \mathbb{R}^n$ given by

$$v_s^z(\gamma) = \left[v_s^1, \ldots, v_s^n\right]^\mathsf{T} = \left[t_1^1 - t_0^1, \quad \ldots, \quad t_1^n - t_0^n\right]^\mathsf{T} \qquad 0 \leq s \leq 1. \tag{5}$$

Informally, we view $v_s^z : \mathcal{T}^n \to \mathbb{R}^n$ as prescribing a constant (but different) velocity to each of the $n$ events. For a fixed pair $z = (\gamma_0, \gamma_1)$ and a given sample $\gamma_0' \sim \eta_0^z$, solving the system of ordinary differential equations $\mathrm{d}\gamma_s' = v_s^z(\gamma_s') \, \mathrm{d}s$ with initial condition $\gamma_0'$ will result in a collection of events which is concentrated around the true event times $\gamma_1$. Note that here, we view this differential equation as an ODE in $\mathcal{T}^n$. If we draw many samples $\gamma_0 \sim \eta_0^z$ and solve the corresponding ODE, the distribution over events at any intermediate time $s$ will be given by $\eta_s^z$.

In other words, the vector field $v_s^z$ generates the path of distributions $\eta_s^z$. However, this path is conditioned on $z$, and we would like to find the vector field $v_s$ which generates the *unconditional* path $\eta_s$. As is standard in flow matching (Lipman et al., 2023; Tong et al., 2024; Albergo & Vanden-Eijnden, 2023), the unconditional vector field $v_s : \mathcal{T}^n \to \mathbb{R}^n$ may be obtained via

$$v_s(\gamma) = \int v_s^z(\gamma) \frac{\mathrm{d}\eta_s^z}{\mathrm{d}\eta_s}(\gamma) \, \mathrm{d}\rho(z). \tag{6}$$

We have thus far described a procedure for interpolating between a given reference distribution $\mu_0^n$ and the data distribution $\mu_1^n$ for a given, fixed number of events $n$. As $n$ was arbitrary, we have successfully constructed a family of interpolants which will enable us to sample from the joint event distribution for any $n$. However, to fully characterize the TPP distribution, we need to also specify the event count distribution. For unconditional generation tasks, this is straightforward – we simply follow the empirical event count distribution see in the training data. We describe our approach for modeling the event count distribution in conditional tasks in the following section.

---

**Algorithm 1:** Training Step for `EventFlow`

1 Sample $\gamma_1 \sim \mu_1$, $s \sim \mathcal{U}[0,1]$, and $\epsilon \sim \mathcal{N}(0,1)$
2 $e_{\mathcal{H}} = \varnothing$                                               `/* Null history */`
3 **if** *forecast* **then**
4      Sample split time $T_0 \in [\Delta T, T - \Delta T]$
5      Construct history $\mathcal{H} \leftarrow \{t \in \gamma_1 : t \leq T_0\}$
6      Embed history $e_{\mathcal{H}} \leftarrow f_\theta(\mathcal{H})$
7      Construct future $\gamma_1 \leftarrow \{t \in \gamma_1 : T_0 < t \leq T_0 + \Delta T\}$
8 Set $n \leftarrow N(\gamma_1)$
9 Sample $t_0^1, \ldots, t_0^n \sim q$ and sort to construct $\gamma_0$
10 Compute $\gamma_s^z$ via $t_s^k = (1-s)t_0^k + st_s^k$
11 Take a gradient step on $\|\gamma_1 - \gamma_0 - v_\theta(\gamma_s^z + \epsilon, s, e_{\mathcal{H}})\|^2$

---

### 4.4 TRAINING, PARAMETRIZATION, AND SAMPLING

To train the model, we would like to perform regression on the vector fields $v_s$ in Equation (6). If we knew this vector field $v_s$, we could draw samples from the data TPP by drawing a sample event sequence $\gamma_0 \sim \mu_0$ from the reference TPP, and flowing each event along the vector field $v_s$.

**Training** Foremost, although the marginal vector field in Equation (6) admits an analytical form, it is intractable to compute in practice as the marginal measure $\eta_s$ is not available. To overcome this, we may instead perform regression on the *conditional* vector fields $v_s^z$. Here, $v_\theta(\gamma_s, s)$ will represent a neural network with parameters $\theta$ which takes in a sequence $\gamma_s$ of $N(\gamma_s) = n$ event times, along with the flow time $s$. That is, we seek to minimize the loss

$$J(\theta) = \mathbb{E}_{s,(\gamma_0,\gamma_1),\hat{\gamma}_s^z}\left[\|\gamma_1 - \gamma_0 - v_\theta(\hat{\gamma}_s^z, s)\|^2\right] \tag{7}$$

which previous works on flow matching have shown to be equal to MSE regression on the *unconditional* $v_s$ up to an additive constant not depending on $\theta$ (Lipman et al., 2023; Tong et al., 2024). We note here that, although the regression target $v_s^z$ is linear, the unconditional vector field $v_s$ will in general be nonlinear. In practice, this loss is estimated by uniformly sampling a flow time $s \in [0,1]$, a pair $z = (\gamma_0, \gamma_1) \sim \rho$ from our balanced coupling and drawing a noisy interpolant $\hat{\gamma}_s^z \sim \eta_s^z$.

To train the model on a forecasting task, where the goal is to predict a future sequence of events conditioned on a history $\mathcal{H}$, we embed $\mathcal{H}$ into a fixed-dimensional vector representation $e_{\mathcal{H}} = f_\theta(\mathcal{H})$ via a learned encoder $f_\theta$ before providing this to the model $v_\theta(\gamma_s, s, e_{\mathcal{H}})$ and minimizing Equation (7). Note that we jointly train the encoder $f_\theta$ and vector field $v_\theta$. See Algorithm 1.

**Parametrization** The second challenge is that we must learn a vector field $v_\theta(\gamma, s)$ in $n$ dimensions for arbitrary values of $n$. In other words, $v_\theta$ is a neural network which takes in a flow time $s \in [0,1]$ and a sequence of events $\gamma$, and must produce $N(\gamma)$ scalar outputs. We achieve this through an attention-based architecture, which we detail in Appendix D. At a high level, the flow time $s$ is transformed via a learnable embedding into a fixed-dimensional vector. Similarly, each event in $\gamma$ is transformed into fixed-dimensional vector via a learned embedding (which is shared across the events, but not the flow time). The flow-time embedding is then added to each event embedding, and the resulting sequence is passed through a standard transformer architecture (Yang et al., 2022; Vaswani, 2017), resulting in a sequence of $N(\gamma)$ vectors. Finally, each of these vectors is projected into one dimension via a linear layer to produce the sequence of $N(\gamma)$ velocities.

For conditional tasks, we must also compute an encoding $e_{\mathcal{H}} = f_\theta(\mathcal{H})$ of the history $\mathcal{H}$. This is done by a separate transformer encoder, which operates in the same fashion as described in the previous paragraph, but without the use of the flow-time $s$ as an input and without the final linear projection layer. This embedding is fed into the intermediate layers of our velocity network via cross-attention.

Lastly, for forecasting tasks we must also learn a model of the event count distribution $p_\phi(n \mid \mathcal{H})$. We treat this as a classification problem, where the goal is to predict the number of events $n$ occurring in the forecast window given the history $\mathcal{H}$. We again use an attention-based model, analogous to our

---

**Algorithm 2:** Sampling Step for `EventFlow`

1  Choose a flow time discretization $0 = s_0 < s_1 < \cdots < s_K = 1$
2  $e_{\mathcal{H}} = \varnothing$                                                          `/* Null history */`
3  **if** *forecast* **then**
4       Embed history $e_{\mathcal{H}} \leftarrow f_\theta(\mathcal{H})$
5       Sample $n \sim p_\phi(n \mid \mathcal{H})$
6  **else**
7       Sample $n \sim \mu_1(n)$
8  Sample $t_0^1, \ldots, t_0^n \sim q$ and sort to construct $\gamma_0$
9  **for** $k = 1, 2, \ldots, K$ **do**
10      $h_k \leftarrow s_k - s_{k-1}$
11      $\gamma_{s_k} \leftarrow \gamma_{s_{k-1}} + h_k v_\theta(\gamma_{s_{k-1}}, s_{k-1}, e_{\mathcal{H}})$

---

velocity model, but where we aggregate the final sequence embedding by averaging and passing this through a small MLP. The model $p_\phi(n \mid \mathcal{H})$ is trained by minimizing the cross-entropy loss.

**Sampling**    Once $v_\theta$ is learned, we may sample from the model by drawing a reference sequence $\gamma_0 \sim \mu_0$ and solving the corresponding ODE parametrized by $v_\theta$. More concretely, we first fix a number of events $n$. When seeking to unconditionally generate new sequences from the underlying data TPP $\mu_1$, we simply sample $n$ from the empirical event count distribution $\mu_1(n)$. For conditional tasks, we draw $n \sim p_\phi(n \mid \mathcal{H})$ from the learned conditional distribution over event counts. Next, we draw $n$ initial events, corresponding to $s = 0$, by sampling and sorting $t_0^1, \ldots, t_0^n \sim q$. In practice, we use $q = \mathcal{N}(0, I_n)$ as we normalize our sequences into the range $[-1, 1]$ during training and sampling (followed by renormalization to the data scale). Since we have fixed $n$, we may view this initial draw as a vector $\gamma_0 = [t_0^1, \ldots, t_0^n] \in \mathcal{T}^n$. This event sequence $\gamma_0$ then serves as the initial condition for the system of ODEs $d\gamma_s = v_\theta(\gamma_s, s) \, ds$ which can be solved numerically. In our experiments, we use the forward Euler scheme, i.e., we specify a discretization $\{0 = s_0 < s_1 < \cdots < s_K = 1\}$ of the flow time (in practice, uniform) and recursively compute

$$\gamma_{s_k} = \gamma_{s_{k-1}} + h_k v_\theta(\gamma_{s_{k-1}}, s_{k-1}) \qquad k = 1, 2, \ldots, K \tag{8}$$

where $h_k = s_k - s_{k-1}$ represents a scalar step size. While other choices of numerical solvers are certainly possible, we found that this simple scheme was sufficient as the model sample paths are typically close to linear. See Algorithm 2 for the full procedure.

## 5   EXPERIMENTS

We study our proposed `EventFlow` model under two settings. The first is a conditional task, where we seek to forecast both the number and times of future events given a history. The second is an unconditional task, where we aim to learn a representation of the underlying TPP distribution from empirical observations and generate new sequences from this distribution. In a sense, this second task can be viewed as a special case of the first with no observed history. Our overall experimental procedure is inspired by that of Lüdke et al. (2023). We evaluate our model across a diverse set of datasets encompassing a wide range of possible point process behaviors. First, we use a collection of six synthetic datasets produced by Omi et al. (2019). We additionally evaluate our model on seven real-world datasets, which are a standard benchmark for modeling unmarked TPPs (Shchur et al., 2020b; Bosser & Taieb, 2023; Lüdke et al., 2023). See Appendix A.

**Baseline Models**    Our baselines were selected as they constitute a set of diverse and highly performant models. For an intensity-based method, we compare against the Neural Hawkes Process (NHP) (Mei & Eisner, 2017). We additionally compare against two intensity-free methods, namely the flow-based IFTPP model (Shchur et al., 2020a) and the diffusion-based model of Lin et al. (2022). Lastly, our strongest baseline is the recently proposed Add-and-Thin model of Lüdke et al. (2023), which can be loosely viewed as a non-autoregressive diffusion model. These models use an RNN-based history encoder, with the exception of Add-and-Thin which uses a CNN-based encoder. See Appendix E for additional details.

Table 1: Sequence distance (9) between the forecasted and ground-truth event sequences on a held-out test set. Lower is better. We report the mean $\pm$ one standard deviation over five random seeds. The best mean distance on each dataset is indicated in bold, and the second best by an underline.

| | PUBG | Reddit-C | Reddit-S | Taxi | Twitter | Yelp-A | Yelp-M |
|---|---|---|---|---|---|---|---|
| IFTPP | $4.2_{\pm0.7}$ | $25.6_{\pm2.3}$ | $61.2_{\pm3.2}$ | $5.1_{\pm0.4}$ | $2.9_{\pm0.2}$ | $2.1_{\pm0.2}$ | $3.4_{\pm0.2}$ |
| NHP | $\underline{2.8}_{\pm0.1}$ | $31.0_{\pm1.4}$ | $95.7_{\pm0.7}$ | $4.5_{\pm0.3}$ | $3.4_{\pm0.5}$ | $\underline{1.8}_{\pm0.1}$ | $\underline{3.0}_{\pm0.2}$ |
| Diffusion | $5.4_{\pm1.2}$ | $25.7_{\pm0.9}$ | $80.3_{\pm11.4}$ | $4.6_{\pm0.7}$ | $\underline{2.4}_{\pm0.2}$ | $\underline{1.8}_{\pm0.1}$ | $3.3_{\pm0.7}$ |
| Add-and-Thin | $\mathbf{2.5}_{\pm0.04}$ | $\mathbf{22.2}_{\pm4.6}$ | $\underline{34.3}_{\pm0.4}$ | $3.7_{\pm0.1}$ | $3.1_{\pm0.2}$ | $\underline{1.8}_{\pm0.1}$ | $\underline{3.0}_{\pm0.2}$ |
| EventFlow (25 NFEs) | $\underline{2.8}_{\pm0.7}$ | $\underline{22.6}_{\pm2.7}$ | $\mathbf{21.5}_{\pm0.4}$ | $3.7_{\pm0.1}$ | $\mathbf{1.7}_{\pm0.1}$ | $\mathbf{1.4}_{\pm0.04}$ | $\mathbf{2.1}_{\pm0.1}$ |
| EventFlow (10 NFEs) | $2.8_{\pm0.7}$ | $22.6_{\pm2.7}$ | $21.3_{\pm0.4}$ | $3.5_{\pm0.2}$ | $1.7_{\pm0.1}$ | $1.4_{\pm0.04}$ | $2.1_{\pm0.1}$ |
| EventFlow (1 NFE) | $2.7_{\pm0.7}$ | $22.6_{\pm2.7}$ | $21.1_{\pm0.3}$ | $3.7_{\pm0.4}$ | $1.8_{\pm0.1}$ | $1.6_{\pm0.2}$ | $2.1_{\pm0.1}$ |
| EventFlow (25 NFEs, true $n$) | $1.2_{\pm0.01}$ | $5.5_{\pm0.3}$ | $8.8_{\pm0.2}$ | $1.8_{\pm0.02}$ | $0.7_{\pm0.01}$ | $0.7_{\pm0.02}$ | $1.1_{\pm0.02}$ |

**Metrics** Evaluating generative TPP models is challenging, as one must take into account both the variable locations and numbers of events. This is particularly challenging for the unconditional setting, where unlike forecasting, we do not have a ground-truth sequence to compare against. Our starting point is a metric (Xiao et al., 2017a) on the space of sequences $\Gamma$, allowing us to measure the distance between two sequences $\gamma = \sum_{k=1}^{n} \delta[t_k^\gamma]$ and $\eta = \sum_{k=1}^{m} \delta[t_k^\eta]$ with possibly different numbers of events. Without loss of generality, we assume $n \leq m$, so the distance is given by

$$d(\gamma, \eta) = \sum_{k=1}^{n} |t_k^\gamma - t_k^\eta| + \sum_{k=n+1}^{m} (T - t_k^\eta) \tag{9}$$

where we recall that sequences are supported on $\mathcal{T} = [0, T]$. This distance can be understood either as an $L^1$ distance between the counting processes of $\gamma, \eta$ or as a generalization of the 1-Wasserstein distance to measures of unequal mass, allowing us to compare two sequences of any lengths.

For our unconditional experiment, we require a metric which will capture the distance between the TPP distributions themselves. To do so we use the distance in Equation (9) to calculate an MMD (Gretton et al., 2012; Shchur et al., 2020b). The MMD between TPPs $\mu, \nu \in \mathbb{P}(\Gamma)$ is given by

$$\text{MMD}(\mu, \nu) = \mathbb{E}_{\gamma, \gamma' \sim \mu}[k(\gamma, \gamma')] - 2\mathbb{E}_{\gamma \sim \mu, \eta \sim \nu}[k(\gamma, \eta)] + \mathbb{E}_{\eta, \eta' \sim \nu}[k(\eta, \eta')] \tag{10}$$

where $k$ is a specified kernel. We use an exponential kernel $k(\gamma, \eta) = \exp\left(-d(\gamma, \eta)/(2\sigma^2)\right)$ with $\sigma$ chosen to be the median distance between all sequences (Shchur et al., 2020b; Lüdke et al., 2023).

## 5.1 FORECASTING EVENT SEQUENCES

We first evaluate our model on a multi-step forecasting task. We set a forecast horizon $\Delta T$ for each of our real-world datasets, and generate event sequences in the range $[T_0, T_0 + \Delta T]$ for some given $T_0$, conditioned on the history of events $\mathcal{H}_{T_0}$. Up to a shift, this means we are taking $\mathcal{T} = [0, \Delta T]$ as the support for our model TPP. The forecast horizon $\Delta T$ is chosen such that the window typically contains multiple events. At training time, we uniformly sample $T_0 \in [\Delta T, T - \Delta T]$ and split a given data sequence $\gamma_1$ into a history on $[0, T_0]$ and a future $[T_0, T_0 + \Delta T]$. We encode the history $\mathcal{H}_{T_0}$ before training the model to fit the events occurring in the future. At testing time, we perform the same splitting procedure, sampling 50 values of $T_0$ for each test set sequence. We then forecast the sequence in $[T_0, T_0 + \Delta T]$ via the model and compute the distance (9) between the ground-truth and generated sequences. Importantly, we note that the distance in Equation (9) is computed using $T_0 + \Delta T$ rather than $T$ as the maximum event time, as using $T$ would result in a distance which is sensitive to the location of the forecasting window within the support $[0, T]$. We further normalize Equation (9) by $\Delta T$ to allow for comparison across different window lengths.

We report the results of this experiment in Table 1. Our proposed `EventFlow` method obtains the lowest average forecasting error on 5/7 of the datasets, and closely matches the performance of Add-and-Thin on the remaining 2/7 datasets. Given that the non-autoregressive models (`EventFlow`, Add-and-Thin) consistently outperform the autoregressive baselines, this is clear evidence that autoregressive models can struggle on multi-step predictions. This is especially true on the Reddit-C and Reddit-S datasets, which exhibit long sequence lengths. In Appendix C, we provide additional evaluations of the event count distributions and one-step prediction performance in terms of MSE.

Table 2: MMDs (1e-2) between the test set and $1,000$ generated sequences averaged over five random seeds. Lower is better. The lowest and second lowest MMD distances are bolded and underlined.

| | H1 | H2 | NSP | NSR | SC | SR | PUBG | Red.C | Red.S | Taxi | Twitter | YelpA | YelpM |
|---|---|---|---|---|---|---|---|---|---|---|---|---|---|
| Data | 1.3 | 1.3 | 1.8 | 3.0 | 5.7 | 1.1 | 1.3 | 0.6 | 0.4 | 3.1 | 2.6 | 3.6 | 3.1 |
| IFTPP | **1.5** | **1.4** | **2.3** | 6.2 | **5.8** | **1.3** | 5.7 | 1.3 | 1.9 | 5.8 | **2.9** | 8.2 | 5.1 |
| NHP | 1.9 | 5.2 | 3.6 | 12.6 | 25.4 | 5.0 | 7.2 | 2.2 | 22.5 | 5.0 | 7.3 | 6.7 | 6.1 |
| Diffusion | 4.8 | 5.5 | 10.8 | 15.0 | 9.1 | 5.1 | 14.3 | 3.9 | 6.2 | 11.7 | 12.5 | 10.9 | 10.5 |
| Add-Thin | 1.9 | 2.5 | 2.6 | 7.4 | 22.5 | 2.2 | 2.8 | 1.2 | 2.7 | 5.2 | 4.8 | **4.5** | **3.0** |
| EventFlow (25 NFEs) | 1.9 | 2.2 | 3.8 | **4.2** | 8.3 | 1.7 | **1.5** | **0.7** | **0.7** | **3.5** | 4.9 | 6.6 | **3.0** |

**Ablations** We additionally perform two ablations. First, we vary the number of function evaluations (NFEs) used at sampling time, i.e., steps in Equation (8). We find that 10 NFEs is sufficient, and increasing the NFEs further does not result in significant gains. Interestingly, with only one step, we observe only a small drop in forecasting performance. This is enabled by our carefully designed interpolant construction (Equation (3)). We emphasize that Add-and-Thin uses 100 NFEs at generation time and the diffusion model uses 1000 NFEs *per generated event*. The autoregressive baselines (NHP, IFTPP) require one NFE per generated event. Thus, our method is able to simultaneously obtain strong forecasting performance while only requiring a small number of model evaluations. In our second ablation, we do not sample $n \sim p_\phi(n \mid \mathcal{H})$, but rather set $n$ to be the true number of events in the forecast window. While this is not practical, this serves to ablate the effect of errors in the event counts. We see that the forecasting error improves significantly, indicating that designing stronger techniques for modeling $p_\phi(n \mid \mathcal{H})$ can lead to improved forecasts.

### 5.2 UNCONDITIONAL GENERATION OF EVENT SEQUENCES

Next, we evaluate our model on an unconditional generation task, where we aim to generate new sequences from the underlying data distribution. This task serves as a benchmark to evaluate the methods in terms of how well they are able to fit the underlying TPP. Moreover, learning a general-purpose TPP prior could enable downstream tasks, such as data augmentation (Graikos et al., 2022). In Table 2 we report MMD values (10) for each of the synthetic and real-world datasets. Model tuning and selection is based on the validation set MMD. MMDs are calculated by sampling 1,000 sequences from each trained model, and estimating Equation (10) using the generated and test set samples. The first row ("data") is the MMD calculated between samples in the training and validation sets, giving us a sense of the best-case performance. See Appendix C for results with standard deviations.

Overall, we find that our EventFlow method (mean rank: $1.8$) exhibits uniformly strong performance, obtaining either the best or second best MMD on 11 of the 13 datasets. This is particularly pronounced on the real-world datasets, where we obtain the lowest MMD on 5 of the 7 datasets. We see that IFTPP (mean rank: $2.1$) is a strong baseline, obtaining results which are competitive with our method. The Add-and-Thin method (mean rank: $2.4$) is often similarly strong, but struggles on the SC dataset. While the NHP (mean rank: $3.7$) can obtain good fits, this appears to be dataset dependent, with weak results on the NSR, SC, and Reddit-S datasets. The diffusion baseline (mean rank: $4.8$) is our weakest baseline, which is perhaps unsurprising as this model can only be trained to maximize the likelihood of a subsequent event and not the overall sequence likelihood.

## 6 CONCLUSION

In this work, we propose EventFlow, a non-autoregressive generative model for temporal point processes. We demonstrate that EventFlow is able to achieve state-of-the-art results on a multi-step forecasting task and strong performance on unconditional generation. There are several directions in which our work could be extended. First, we do not explicitly enforce that the support of our model TPP is $[0, T]$, which would necessitate moving beyond the Gaussian setting. Second, more sophisticated approaches to learning the event count distribution $p_\phi(n \mid \mathcal{H})$ could lead to improved performance. Our work lays a foundation for flow-based TPPs, and exploring applications in tasks like imputation, marked TPPs, and querying (Boyd et al., 2023) are exciting directions.

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

## A  DATASETS

In this section, we provide some additional details regarding the datasets used in this work. In Table 3, we report the number of sequences in each dataset, some basic statistics regarding the number of events in each sequence, and their support $[0, T]$ and chosen forecast window $\Delta T$. In all datasets, we use 60% of the data for training, 20% for validation, and the remaining 20% for testing.

**Synthetic Datasets**   Our synthetic datasets are adopted from those proposed by Omi et al. (2019). Each of these datasets consists of $1,000$ sequences supported on $\mathcal{T} = [0, 100]$. These synthetic datasets are chosen as they exhibit a wide range of behavior, ranging from i.i.d. inter-arrival times to self-correcting processes which discourage rapid bursts of events. We refer to Section 4 of Omi et al. (2019) for details.

**Real-World Datasets**   We use the set of real-world datasets proposed in Shchur et al. (2020b), which constitute a set of standard benchmark datasets for unmarked TPPs. We refer to Appendix D of Shchur et al. (2020b) for additional details. With the exception of PUBG, these datasets are supported on $\mathcal{T} = [0, 24]$, i.e. each sequence corresponds to a single day. For the PUBG dataset, $\mathcal{T} = [0, 38]$ corresponds to the maximum length (in minutes) of an online game of PUBG. We note that PUBG has the largest number of sequences (which can lead to slow training), and the Reddit-C and Reddit-S datasets have long sequences (which can lead to slow training and high memory costs).

Table 3: Some basic summary statistics of the datasets we consider in this work.

| | Sequences | Mean length | Std length | Range length | Support | $\Delta T$ |
|---|---|---|---|---|---|---|
| Hawkes1 | 1000 | 95.4 | 45.8 | [14, 300] | [0, 100] | − |
| Hawkes2 | 1000 | 97.2 | 49.1 | [18, 355] | [0, 100] | − |
| Nonstationary Poisson | 1000 | 100.3 | 9.8 | [71, 134] | [0, 100] | − |
| Nonstationary Renewal | 1000 | 98 | 2.9 | [86, 100] | [0, 100] | − |
| Stationary Renewal | 1000 | 109.2 | 38.1 | [1, 219] | [0, 100] | − |
| Self-Correcting | 1000 | 100.3 | 0.74 | [98, 102] | [0, 100] | − |
| PUBG | 3001 | 76.5 | 8.8 | [26, 97] | [0, 38] | 5 |
| Reddit-C | 1356 | 295.7 | 317.9 | [1, 2137] | [0, 24] | 4 |
| Reddit-S | 1094 | 1129 | 359.5 | [363, 2658] | [0, 24] | 4 |
| Taxi | 182 | 98.4 | 20 | [12, 140] | [0, 24] | 4 |
| Twitter | 2019 | 14.9 | 14 | [1, 169] | [0, 24] | 4 |
| Yelp-Airport | 319 | 30.5 | 7.5 | [9, 55] | [0, 24] | 4 |
| Yelp-Miss. | 319 | 55.2 | 15.9 | [3, 107] | [0, 24] | 4 |

## B  PROOFS

**Proposition 2** (Existence of Balanced Couplings)**.**
*Let $\mu, \nu \in \mathbb{P}(\Gamma)$ be two TPPs. The set of balanced couplings $\Pi_b(\mu, \nu)$ is nonempty if and only if $\mu(n) = \nu(n)$ have the same distribution over event counts.*

*Proof.* Let $A_1, A_2 \subseteq \Gamma$ be Borel measurable (Daley & Vere-Jones, 2003, Prop. 5.3) subsets of the configuration space $\Gamma$, i.e. each of $A_1, A_2$ is a measurable collection of event sequences. Observe that for $i = 1, 2$, each $A_i$ can be written as a disjoint union

$$A_i^n = \bigcup_{n=0}^{\infty} \mathcal{T}^n \cap A_i \tag{11}$$

i.e. $A_i^n \subseteq A_i$ is the subset of $A_i$ containing only sequences with $n$ events. Note each $A_i^n$ is a Borel measurable subset of $\mathcal{T}^n$.

Now, suppose that $\mu(n) = \nu(n)$ have equal event count distributions. We define the coupling $\rho \in \mathbb{P}(\Gamma \times \Gamma)$ by

$$\rho(A_1 \times A_2) = \sum_{n=0}^{\infty} \mu(n) \mu^n(A_1^n) \nu^n(A_2^n). \tag{12}$$

Here, in a slight abuse of notation, we use $\mu^n, \nu^n$ to denote the corresponding joint probability measures over $n$ events, i.e., both are Borel probability measures on $\mathcal{T}^n$. Since the $n$-dimensional projection of $\Gamma$ in Equation (11) is simply $\mathcal{T}^n$, it is immediate that $\rho(A_1 \times \Gamma) = \mu(A_1)$ and $\rho(\Gamma \times A_2) = \nu(A_2)$, so that $\rho$ is indeed a coupling. Moreover, it is clear that the coupling is balanced.

Conversely, suppose $\rho \in \Pi_b(\mu_0, \mu_1)$ is a balanced coupling. Let $N : \Gamma \to \mathbb{Z}_{\geq 0}$ be the event counting functional and let $\pi^1, \pi^2 : \Gamma \times \Gamma \to \Gamma$ denote the canonical projections of $\Gamma \times \Gamma$ onto its components. That is, $\pi^1 : (\gamma_0, \gamma_1) \mapsto \gamma_0$ and $\pi^2 : (\gamma_0, \gamma_1) \mapsto \gamma_1$. Furthermore, let $(N, N) : \Gamma \times \Gamma \to \mathbb{Z}_{\geq 0} \times \mathbb{Z}_{\geq 0}$ denote the product of the counting functional, i.e. $(N, N)(\gamma_0, \gamma_1) = (N(\gamma_0), N(\gamma_1))$. Note that the pushforward $N_{\#}\mu$ yields the event count distribution $\mu(n)$ of $\mu$ (and analogously for $\nu$).

Now, observe that composing the projections and counting functionals yields

$$\pi^1 \circ (N, N) = N \circ \pi^1 \qquad \pi^2 \circ (N, N) = N \circ \pi^2. \tag{13}$$

As $\rho$ is a coupling, we have that $\mu = \pi^1_{\#}\rho$ and $\nu = \pi^2_{\#}\rho$. From these observations, it follows that

$$N_{\#}\mu = N_{\#}\left(\pi^1_{\#}\rho\right) \tag{14}$$

$$= (N \circ \pi^1)_{\#}\rho \tag{15}$$

$$= (\pi^1 \circ (N, N))_{\#}\rho \tag{16}$$

$$= \pi^1_{\#}\left((N, N)_{\#}\rho\right) \tag{17}$$

$$= \pi^2_{\#}\left((N, N)_{\#}\rho\right) \tag{18}$$

$$= N_{\#}\nu \tag{19}$$

where the equality in the penultimate line follows from the fact that $\rho$ is balanced. Thus, we have shown that the existence of a balanced coupling implies that $N_{\#}\mu = N_{\#}\nu$, i.e. the event count distributions are equal. $\qquad\square$

# C  ADDITIONAL EXPERIMENTS

This section contains additional empirical evaluations of our proposed method. First, in Tables 4 and 5, we report the MMD values appearing in the unconditional experiment (i.e., Table 2 in the main paper) with standard deviations. These are omitted from the main paper for the sake of space.

Next, we provide additional evaluations on our forecasting experiment, where we follow the same training and generation procedure described in Section 5.1. In Table 6, we evaluate the performance of the various models only in terms of the predicted number of events in the forecast. To do so, we measure the mean absolute relative error (MARE) given by

$$\text{MARE} = \mathbb{E}_{n,\hat{n}} \left| \frac{\hat{n} - n}{n} \right| \tag{20}$$

where $n$ represents the true number of points in a sequence, $\hat{n}$ represents the predicted number of points, and the expectation is estimated empirically on the testing set. As our method directly predicts the number of events $n$ by sampling from the learned distribution $p_\phi(n \mid \mathcal{H})$, this serves as a direct evaluation of this component of our model. Here, we find that Add-and-Thin has strong performance (mean rank: 1.3), whereas our method (mean rank: 3), diffusion (mean rank: 3.1) perform comparably, while IFTPP (mean rank: 3.6) and NHP lag slightly behind (mean rank: 4). While our method has room for improvement, we note that even though our approach to learning $p(n \mid H)$ is quite simple it still achieves competitive results. Designing better techniques for predicting the event counts is an exciting direction for future work. However, we emphasize that our model shows clear gains on the forecasting metric (Table 1) which measures both the event counts and their times, and this is the primary relevant metric for the problem we address in this paper.

In Table 7, we evaluate the performance of our model when forecasting only a single subsequent event. That is, given a history $\mathcal{H}$, we evaluate the MSE between the first true event time following this history and the first event time generated by each model conditioned on $\mathcal{H}$. The results are reported in Table 7. Generally, all of the methods show similar results on this metric, despite there being clear differences between methods on the multi-step task. We believe this serves to further highlight the necessity of moving beyond single-step prediction tasks.

Table 4: MMDs (1e-2) between the test set and $1,000$ generated sequences on our real-world datasets. Lower is better. We report the mean $\pm$ one standard deviation over five random seeds. The lowest MMD distance on each dataset is indicated in bold, and the second lowest is indicated by an underline.

|  | PUBG | Reddit-C | Reddit-S | Taxi | Twitter | Yelp-A | Yelp-M |
|---|---|---|---|---|---|---|---|
| Data | 1.3 | 0.6 | 0.4 | 3.1 | 2.6 | 3.6 | 3.1 |
| IFTPP | $5.7_{\pm1.8}$ | $1.3_{\pm1.2}$ | $\underline{1.9}_{\pm0.6}$ | $5.8_{\pm0.9}$ | $\mathbf{2.9}_{\pm0.6}$ | $8.2_{\pm4.7}$ | $\underline{5.1}_{\pm0.7}$ |
| NHP | $7.2_{\pm0.2}$ | $2.2_{\pm1.6}$ | $22.5_{\pm0.3}$ | $\underline{5.0}_{\pm0.1}$ | $7.3_{\pm0.7}$ | $6.7_{\pm1.5}$ | $6.1_{\pm2.3}$ |
| Diffusion | $14.3_{\pm6.5}$ | $3.9_{\pm1.2}$ | $6.2_{\pm3.3}$ | $11.7_{\pm1.8}$ | $12.5_{\pm1.9}$ | $10.9_{\pm3.8}$ | $10.5_{\pm5.2}$ |
| Add-and-Thin | $\underline{2.8}_{\pm0.5}$ | $\underline{1.2}_{\pm0.27}$ | $2.7_{\pm0.3}$ | $5.2_{\pm0.6}$ | $\underline{4.8}_{\pm0.4}$ | $\mathbf{4.5}_{\pm0.2}$ | $\mathbf{3.0}_{\pm0.5}$ |
| EventFlow (ours) | $\mathbf{1.5}_{\pm0.2}$ | $\mathbf{0.7}_{\pm0.1}$ | $\mathbf{0.7}_{\pm0.1}$ | $\mathbf{3.5}_{\pm0.1}$ | $4.9_{\pm0.7}$ | $\underline{6.6}_{\pm1.2}$ | $\mathbf{3.0}_{\pm0.5}$ |

Table 5: MMDs (1e-2) between the test set and $1,000$ generated sequences on our synthetic datasets. Lower is better. We report the mean $\pm$ one standard deviation over five random seeds. The lowest MMD distance on each dataset is indicated in bold, and the second lowest is indicated by an underline.

|  | Hawkes1 | Hawkes2 | NSP | NSR | SC | SR |
|---|---|---|---|---|---|---|
| Data | 1.3 | 1.3 | 1.8 | 3.0 | 5.7 | 1.1 |
| IFTPP | $\mathbf{1.5}_{\pm0.4}$ | $\mathbf{1.4}_{\pm0.5}$ | $\mathbf{2.3}_{\pm0.7}$ | $\underline{6.2}_{\pm2.1}$ | $\mathbf{5.8}_{\pm0.5}$ | $\mathbf{1.3}_{\pm0.3}$ |
| NHP | $\underline{1.9}_{\pm0.3}$ | $5.2_{\pm1.6}$ | $3.6_{\pm1.3}$ | $12.6_{\pm1.8}$ | $25.4_{\pm11.5}$ | $5.0_{\pm0.7}$ |
| Diffusion | $4.8_{\pm2.7}$ | $5.5_{\pm3.3}$ | $10.8_{\pm7.5}$ | $15.0_{\pm3.6}$ | $9.1_{\pm1.8}$ | $5.1_{\pm2.8}$ |
| Add-and-Thin | $\underline{1.9}_{\pm0.5}$ | $2.5_{\pm0.3}$ | $\underline{2.6}_{\pm0.5}$ | $7.4_{\pm1.2}$ | $22.5_{\pm0.5}$ | $2.2_{\pm0.8}$ |
| EventFlow (ours) | $\underline{1.9}_{\pm0.2}$ | $\underline{2.2}_{\pm0.1}$ | $3.8_{\pm1.2}$ | $\mathbf{4.2}_{\pm0.5}$ | $\underline{8.3}_{\pm0.4}$ | $\underline{1.7}_{\pm0.3}$ |

Table 6: MARE values evaluating the predicted number of events when forecasting. Mean values $\pm$ one standard deviation are reported over five random seeds. The lowest MARE on each dataset is indicated and bold, and the second lowest is indicated by an underline.

|  | PUBG | Reddit-C | Reddit-S | Taxi | Twitter | Yelp-A | Yelp-M |
|---|---|---|---|---|---|---|---|
| IFTPP | $1.05_{\pm0.14}$ | $1.69_{\pm0.39}$ | $0.79_{\pm0.20}$ | $0.60_{\pm0.11}$ | $0.88_{\pm0.08}$ | $0.76_{\pm0.07}$ | $0.76_{\pm0.05}$ |
| NHP | $1.02_{\pm0.08}$ | $\mathbf{0.95}_{\pm0.01}$ | $1.00_{\pm0.0004}$ | $0.67_{\pm0.11}$ | $2.48_{\pm0.40}$ | $0.80_{\pm0.22}$ | $1.07_{\pm0.34}$ |
| Diffusion | $1.95_{\pm0.48}$ | $1.28_{\pm0.09}$ | $1.12_{\pm0.56}$ | $0.49_{\pm0.07}$ | $\underline{0.66}_{\pm0.04}$ | $0.65 \pm_{0.07}$ | $\underline{0.72}_{\pm0.07}$ |
| Add-and-Thin | $\mathbf{0.43}_{\pm0.01}$ | $\underline{0.99}_{\pm0.10}$ | $\underline{0.38}_{\pm0.01}$ | $\mathbf{0.33}_{\pm0.02}$ | $\mathbf{0.60}_{\pm0.02}$ | $\mathbf{0.42}_{\pm0.01}$ | $\mathbf{0.46}_{\pm0.03}$ |
| Ours | $\underline{0.69}_{\pm0.17}$ | $2.01_{\pm0.40}$ | $\mathbf{0.26}_{\pm0.01}$ | $\underline{0.47}_{\pm0.03}$ | $1.23_{\pm0.07}$ | $\underline{0.66}_{\pm0.03}$ | $0.80_{\pm0.05}$ |

Table 7: MSE values evaluating one-step-ahead forecasting performance. Mean values $\pm$ one standard deviation are reported over five random seeds. The lowest MSE on each dataset is indicated and bold, and the second lowest is indicated by an underline.

|  | PUBG | Reddit-C | Reddit-S | Taxi | Twitter | Yelp-A | Yelp-M |
|---|---|---|---|---|---|---|---|
| IFTPP | $0.85_{\pm0.05}$ | $\underline{0.32}_{\pm0.03}$ | $0.0047_{\pm0.0006}$ | $0.22_{\pm0.03}$ | $1.74_{\pm0.10}$ | $1.24_{\pm0.16}$ | $1.11_{\pm0.17}$ |
| NHP | $0.89_{\pm0.09}$ | $0.53_{\pm0.24}$ | $\mathbf{0.0022}_{\pm0.0007}$ | $0.31_{\pm0.12}$ | $2.00_{\pm0.30}$ | $1.30_{\pm0.26}$ | $\underline{1.03}_{\pm0.35}$ |
| Diffusion | $\mathbf{0.61}_{\pm0.10}$ | $0.33_{\pm0.04}$ | $\underline{0.0037}_{\pm0.0012}$ | $0.23_{\pm0.14}$ | $\mathbf{1.30}_{\pm0.21}$ | $\mathbf{0.86}_{\pm0.18}$ | $\mathbf{0.92}_{\pm0.20}$ |
| Add-and-Thin | $0.86_{\pm0.05}$ | $\mathbf{0.30}_{\pm0.04}$ | $0.0043_{\pm0.0007}$ | $\underline{0.21}_{\pm0.03}$ | $\underline{1.53}_{\pm0.14}$ | $1.16_{\pm0.16}$ | $1.20_{\pm0.14}$ |
| Ours (25 NFEs) | $\underline{0.75}_{\pm0.08}$ | $0.62_{\pm0.09}$ | $0.0137_{\pm0.0015}$ | $\mathbf{0.17}_{\pm0.02}$ | $2.00_{\pm0.08}$ | $\underline{0.95}_{\pm0.07}$ | $1.05_{\pm0.02}$ |
| Ours (10 NFEs) | $0.69_{\pm0.07}$ | $0.59_{\pm0.09}$ | $0.0113_{\pm0.0017}$ | $0.15_{\pm0.01}$ | $1.76_{\pm0.07}$ | $0.81_{\pm0.08}$ | $0.94_{\pm0.03}$ |
| Ours (1 NFE) | $0.64_{\pm0.16}$ | $0.82_{\pm0.32}$ | $0.0472_{\pm0.0098}$ | $0.17_{\pm0.03}$ | $1.40_{\pm0.09}$ | $0.83_{\pm0.09}$ | $0.88_{\pm0.18}$ |

## D   EVENTFLOW ARCHITECTURE AND TRAINING DETAILS

Here, we provide additional details regarding the parametrization and training of our `EventFlow` model. In general, our model is based on the transformer architecture (Vaswani, 2017; Yang et al., 2022), due to its general ability to handle variable length inputs and outputs, high flexibility, and ability to incorporate long-range interactions. In all settings, our reference measure $\mu_0$ is specified with $q = \mathcal{N}(0, I)$.

**Model Parametrization**  For our unconditional model, we first embed the sequence times $\gamma_s$, the flow-time $s$, and the sequence position indices using sinusoidal embeddings followed by an additional linear layer. There are three linear layers in total – one for the flow time, one shared across the sequence times, and one for the position indices. These embeddings are added together to create a representation of the sequence, and we apply a standard transformer to this sequence to produce a sequence of vectors of length $N(\gamma_s)$. Finally, each of these vectors is projected to one dimension via a final linear layer with shared weights to produce the vector field $v_\theta(\gamma_s, s)$. See Figure 2.

For the conditional model, we use a standard transformer encoder-decoder architecture. We first embed the history sequence times $\mathcal{H}$ and the sequence position indices in a manner analogous to the above. In addition, the model was provided the start of the prediction window $T_0$ by concatenating it as the final event in $\mathcal{H}$. This yielded better results than encoding the start of the prediction window separately. We feed these embeddings through the transformer encoder produce an intermediate representation $e_\mathcal{H}$.

For the decoder, we provide the model with the current state $\gamma_s$ (corresponding to the generated event times at flow-time $s$), the flow-time $s$, and the corresponding positional indices. These are embedded as previously described, before being passed into the transformer decoder. The history encoding $e_\mathcal{H}$ is provided to the decoder via cross-attention in the intermediate layer. This produces a sequence of $N(\gamma_s)$ vectors, which we again pass through a final linear layer to produce the final conditional vector field $v_\theta(\gamma_s, s, e_\mathcal{H})$. See Figure 3.

Our architecture for predicting the number of future events given a history, i.e. $p(n \mid \mathcal{H})$, is again based on the transformer decoder, sharing the same overall architecture as our unconditional model. However, the key difference is that we instead take a mean of the final sequence embeddings before passing this through a small MLP to produce the final logit. See Figure 4.

**Training and Tuning**  We normalize all sequences to the range $[-1, 1]$, using the overall min/max event time seen in the training data. All sequences are generated on this normalized scale, prior to re-scaling the sequence back to the original data range before evaluation. Our model is trained with the Adam (Kingma & Ba, 2015) optimizer with $\beta_1 = 0.9$ and $\beta_2 = 0.999$ for $30,000$ steps with a cosine scheduler (Loshchilov & Hutter, 2017), which cycled every $10,000$ steps. Final hyperparameters were selected by best performance on the validation dataset achieved at any point during the training, where models were evaluated 10 times throughout their training.

To tune our model, we performed a grid search over learning rates in $\{5 \times 10^{-3}, 10^{-3}, 5 \times 10^{-4}\}$ and dropout probabilities in $\{0, 0.1, 0.2\}$. Overall, we found that learning rates of $10^{-2}$ or larger often caused the model to diverge, and a dropout of $0.1$ yielded the best results across all settings. We use 6 transformer layers, 8 attention heads, and an embedding dimension of $512$ across all settings, except for the Reddit-C and Reddit-S datasets where we use 4 heads and an embedding dimension of $128$ due to the increased memory cost of these datasets.

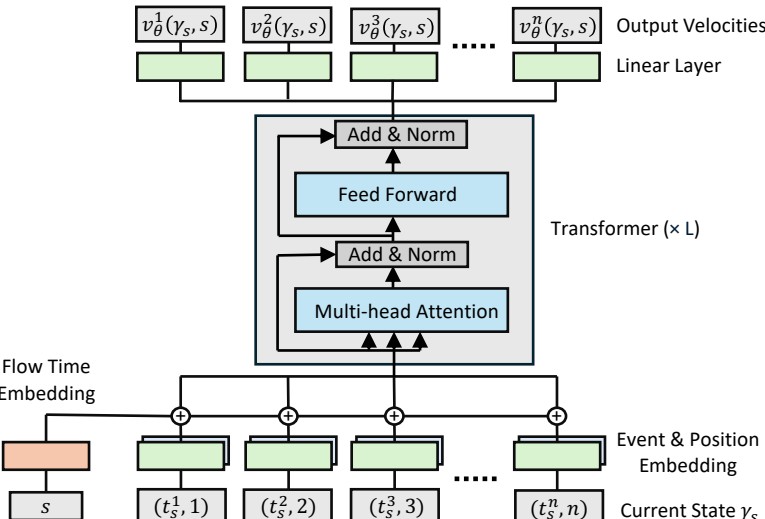

Figure 2: Overview of our model architecture for unconditional generation. The model takes as input the flow time $s$ and current sequence state $\gamma_s = \sum_{k=1}^{n} \delta[t_s^k]$. Each input is projected to a fixed-length vector via a learnable embedding. The resulting embeddings are added and passed to the transformer model, which produces a sequence of output velocities $v_\theta(\gamma_s, s)$ with $N(\gamma_s)$ components.

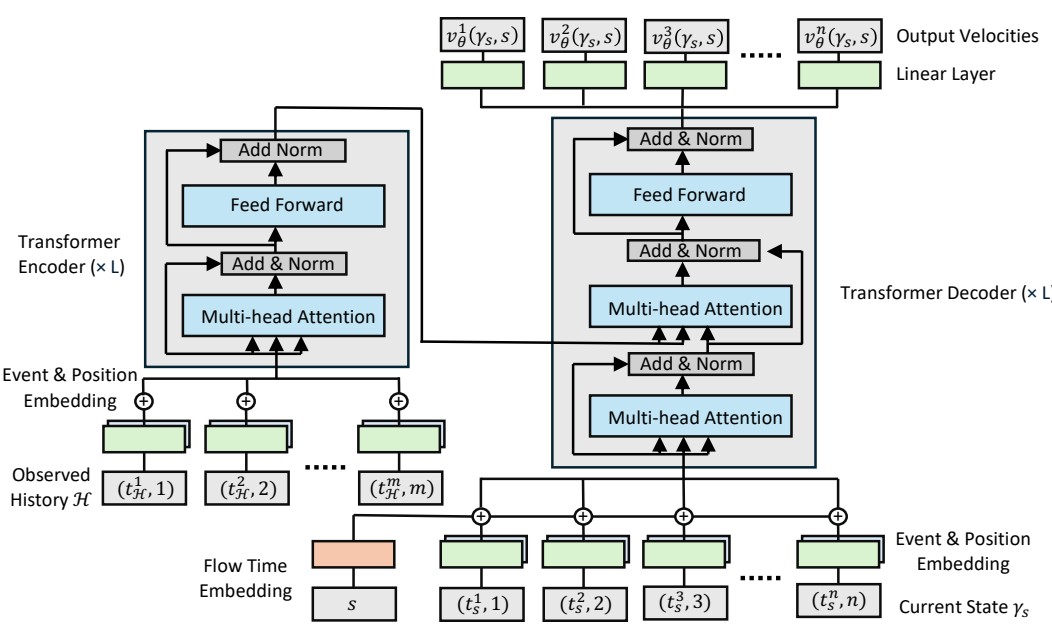

Figure 3: Overview of our model architecture for conditional generation. The encoder (left) takes as input the observed history $\mathcal{H}$, which is embedded in a fashion analogous to our unconditional model. The decoder (right) takes as input the flow time $s$ and current state $\gamma_s = \sum_{k=1} \delta[t_s^k]$. These are embedded and passed through the decoder, which applies cross attention to produce the conditional velocities $v_\theta(\gamma_s, s, e_{\mathcal{H}})$.

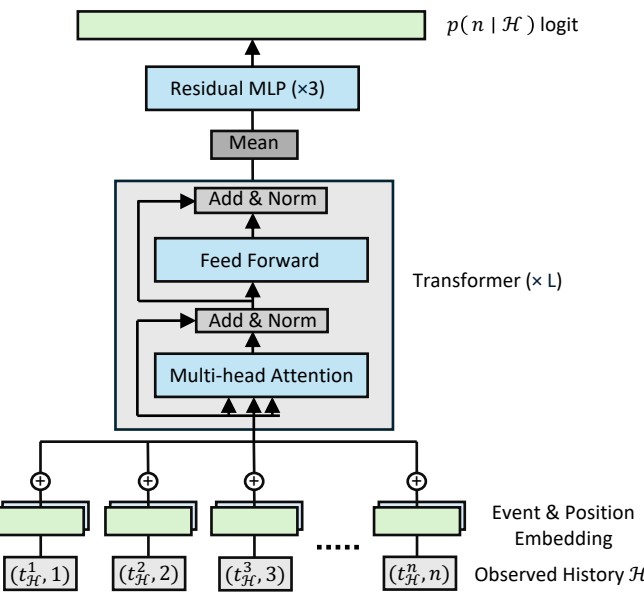

Figure 4: Overview of our architecture modeling the event count distribution $p_\phi(n \mid \mathcal{H})$. The model takes as input an observed history $\mathcal{H}$. As in our other architectures, the events are embedded and passed through a transformer. Here, the final sequence embedding output by the transformer is averaged and passed through an additional residual MLP with three layers to produce the logit corresponding to $p(n \mid \mathcal{H})$.

Table 8: The best hyperparameter settings found for the vector field $v_\theta$ in our EventFlow method on the unconditional generation task.

|  | Learning Rate | Emb. Dim. | MLP Dim | Heads | Transformer Layers |
|---|---|---|---|---|---|
| Hawkes1 | $10^{-3}$ | 512 | 2048 | 8 | 6 |
| Hawkes2 | $10^{-3}$ | 512 | 2048 | 8 | 6 |
| Nonstationary Poisson | $10^{-3}$ | 512 | 2048 | 8 | 6 |
| Nonstationary Renewal | $10^{-3}$ | 512 | 2048 | 8 | 6 |
| Stationary Renewal | $10^{-3}$ | 512 | 2048 | 8 | 6 |
| Self-Correcting | $10^{-3}$ | 512 | 2048 | 8 | 6 |
| PUBG | $5 \times 10^{-4}$ | 512 | 2048 | 8 | 6 |
| Reddit-C | $10^{-3}$ | 128 | 256 | 4 | 6 |
| Reddit-S | $5 \times 10^{-3}$ | 128 | 256 | 4 | 6 |
| Taxi | $5 \times 10^{-4}$ | 512 | 2048 | 8 | 6 |
| Twitter | $10^{-3}$ | 512 | 2048 | 8 | 6 |
| Yelp-Airport | $5 \times 10^{-4}$ | 512 | 2048 | 8 | 6 |
| Yelp-Miss. | $10^{-3}$ | 512 | 2048 | 8 | 6 |

Table 9: The best hyperparameter settings found for the vector field $v_\theta$ in our EventFlow method on the forecasting task.

|  | Learning Rate | Emb. Dim. | MLP Dim. | Heads | Transformer Layers |
|---|---|---|---|---|---|
| PUBG | $10^{-3}$ | 512 | 2048 | 8 | 6 |
| Reddit-C | $10^{-3}$ | 128 | 256 | 4 | 6 |
| Reddit-S | $10^{-3}$ | 128 | 256 | 4 | 6 |
| Taxi | $10^{-3}$ | 512 | 2048 | 8 | 6 |
| Twitter | $5 \times 10^{-4}$ | 512 | 2048 | 8 | 6 |
| Yelp-Airport | $10^{-3}$ | 512 | 2048 | 8 | 6 |
| Yelp-Miss. | $10^{-3}$ | 512 | 2048 | 8 | 6 |

Table 10: The best hyperparameter settings found for the event count predictor $p(n \mid \mathcal{H})$ in our EventFlow method on the forecasting task.

|  | Learning Rate | Emb. Dim. | MLP Dim. | Heads | Transformer Layers |
|---|---|---|---|---|---|
| PUBG | $5 \times 10^{-4}$ | 512 | 2048 | 8 | 6 |
| Reddit-C | $10^{-3}$ | 128 | 256 | 4 | 6 |
| Reddit-S | $10^{-3}$ | 128 | 256 | 4 | 6 |
| Taxi | $5 \times 10^{-4}$ | 512 | 2048 | 8 | 6 |
| Twitter | $5 \times 10^{-4}$ | 512 | 2048 | 8 | 6 |
| Yelp-Airport | $5 \times 10^{-4}$ | 512 | 2048 | 8 | 6 |
| Yelp-Miss. | $5 \times 10^{-4}$ | 512 | 2048 | 8 | 6 |

# E    ADDITIONAL DETAILS ON BASELINES

In this section, we provide additional details regarding our baseline methods. All methods are trained at a batch size of $64$ for $1,000$ epochs, using early stopping on the validation set loss. In early experiments, we also evaluated AttNHP (Zuo et al., 2020), a variant of the NHP which uses an attention-based encoder, but found it to be prohibitively expensive in terms of memory cost (requiring more than 24 GB of VRAM) and, as a result, do not include it in our results.

**IFTPP**    Our first baseline is the intensity-free TPP model of Shchur et al. (2020a). This model uses an RNN encoder and a mixture of log-normal distributions to parametrize the decoder. We directly use the implementation provided by the authors.[2] We train for $1,000$ epochs with early stopping based on the validation set loss. To tune this baseline, we performed a grid search over learning rates in $\{10^{-4}, 10^{-3}, 10^{-2}\}$, weight decays in $\{0, 10^{-6}, 10^{-5}, 10^{-4}\}$, history embedding dimensions $\{32, 64, 128\}$, and mixture component counts $\{8, 16, 32, 64\}$. Our best hyperparameters can be found in Table 11 and Table 12.

Table 11: The best hyperparameter settings found for IFTPP on the unconditional generation task.

|  | Learning Rate | Weight Decay | Embedding Dimension | Mixture Components |
|---|---|---|---|---|
| Hawkes1 | $10^{-3}$ | $10^{-4}$ | 32 | 8 |
| Hawkes2 | $10^{-2}$ | 0 | 32 | 8 |
| Nonstationary Poisson | $10^{-3}$ | $10^{-6}$ | 128 | 8 |
| Nonstationary Renewal | $10^{-2}$ | $10^{-6}$ | 64 | 16 |
| Stationary Renewal | $10^{-3}$ | $10^{-4}$ | 32 | 8 |
| Self-Correcting | $10^{-3}$ | $10^{-6}$ | 32 | 64 |
| PUBG | $10^{-2}$ | 0 | 128 | 32 |
| Reddit-C | $10^{-3}$ | $10^{-4}$ | 64 | 16 |
| Reddit-S | $10^{-2}$ | $10^{-4}$ | 64 | 16 |
| Taxi | $10^{-2}$ | $10^{-5}$ | 128 | 64 |
| Twitter | $10^{-3}$ | $10^{-4}$ | 64 | 6 |
| Yelp-Airport | $10^{-2}$ | $10^{-6}$ | 64 | 64 |
| Yelp-Miss. | $10^{-3}$ | $10^{-4}$ | 32 | 8 |

Table 12: The best hyperparameter settings found for IFTPP on the forecasting task.

|  | Learning Rate | Weight Decay | Embedding Dimension | Mixture Components |
|---|---|---|---|---|
| PUBG | $10^{-4}$ | $10^{-6}$ | 32 | 32 |
| Reddit-C | $10^{-2}$ | 0 | 64 | 8 |
| Reddit-S | $10^{-2}$ | 0 | 64 | 16 |
| Taxi | $10^{-3}$ | $10^{-6}$ | 128 | 8 |
| Twitter | $10^{-2}$ | $10^{-5}$ | 32 | 8 |
| Yelp-Airport | $10^{-2}$ | $10^{-6}$ | 128 | 32 |
| Yelp-Miss. | $10^{-2}$ | $10^{-6}$ | 32 | 8 |

**NHP**    We additionally compare against the Neural Hawkes Process of Mei & Eisner (2017). This model uses an LSTM encoder and a parametric form, whose weights are modeled by a neural network, to model the conditional intensity function. In practice, we use the implementation proved by the EasyTPP benchmark (Xue et al., 2024), as this version implements the necessary thinning algorithm for sampling.[3] We perform a grid search over learning rates in $\{10^{-4}, 10^{-3}, 10^{-2}\}$ and embedding dimensions in $\{32, 64, 128\}$. These hyperparameters are chosen as the EasyTPP implementation allows these to be configured easily. Our best hyperparameters are reported in Table 13 and Table 14.

---

[2]URL: https://github.com/shchur/ifl-tpp
[3]URL: https://github.com/ant-research/EasyTemporalPointProcess

**Diffusion**  Our diffusion baseline is based on the implementation of Lin et al. (2022), and our decoder model architecture is taken directly from the code of Lin et al. (2022).[4] At a high level, this model is a discrete-time diffusion model (Ho et al., 2020) trained to generate a single inter-arrival time given a history embedding. Note that as the likelihood is not available in diffusion models, the CDF in the likelihood in Equation (2) is not tractable. Instead, the model is trained by maximizing an ELBO of only the subsequent inter-arrival time.

In preliminary experiments, we found that the codebase provided by Lin et al. (2022) often produced `NaN` values during sampling, prompting us to make several changes. First, we use the RNN encoder from Shchur et al. (2020a), i.e. the same encoder as the IFTPP baseline, to reduce the memory requirements of the model. Second, we do not log-scale the inter-arrival times as suggested by Lin et al. (2022), as we found that this often led to overflow and underflow issues at sampling time. Third, we do not normalize the data via standardization (i.e., subtracting off the mean inter-arrival time and dividing by the standard deviation), but rather, we scale the inter-arrival times so that they are in the bounded range $[-1, 1]$. This is aligned with standard diffusion implementations (Ho et al., 2020), and allows us to perform clipping at sampling time to avoid the accumulation of errors. With these changes, our diffusion baseline is competitive, and able to obtain stronger results than previous work has reported (Lüdke et al., 2023).

We use 1000 diffusion steps and the cosine beta schedule (Nichol & Dhariwal, 2021), and we train the model on the simplified $\epsilon$-prediction loss of Ho et al. (2020). We train for $1,000$ epochs with early stopping based on the validation set loss. To tune this baseline, we performed a grid search over learning rates in $\{10^{-4}, 10^{-3}, 10^{-2}\}$, weight decays in $\{0, 10^{-6}, 10^{-5}, 10^{-4}\}$, history embedding dimensions $\{32, 64, 128\}$, and layer numbers $\{2, 4, 6\}$. Our best hyperparameters can be found in Table 15 and Table 16.

**Add-and-Thin**  We compare to the `Add-and-Thin` model of Lüdke et al. (2023) as a recently proposed non-autoregressive baseline. We directly run the code provided by the authors without additional modifications.[5] We do, however, perform a slightly larger hyperparameter sweep than Lüdke et al. (2023), in order to ensure a fair comparison between the methods considered. We train for $1,000$ epochs with early stopping on the validation loss. Tuning is performed via a grid search over learning rates in $\{10^{-4}, 10^{-3}, 10^{-2}\}$ and number of mixture components in $\{8, 16, 32, 64\}$. We choose to tune only these hyperparameters in order to follow the implementation provided by the authors. Our best hyperparameters can be found in Table 17 and Table 18.

---

[4] URL: `https://github.com/EDAPINENUT/GNTPP`
[5] URL: `https://github.com/davecasp/add-thin`

Table 13: The best hyperparameter settings found for NHP on the unconditional generation task.

|  | Learning Rate | Embedding Dimension |
|---|---|---|
| Hawkes1 | $10^{-3}$ | 64 |
| Hawkes2 | $10^{-3}$ | 64 |
| Nonstationary Poisson | $10^{-3}$ | 64 |
| Nonstationary Renewal | $10^{-4}$ | 64 |
| Stationary Renewal | $10^{-3}$ | 64 |
| Self-Correcting | $10^{-3}$ | 64 |
| PUBG | $10^{-4}$ | 64 |
| Reddit-C | $10^{-2}$ | 64 |
| Reddit-S | $10^{-2}$ | 64 |
| Taxi | $10^{-2}$ | 64 |
| Twitter | $10^{-4}$ | 64 |
| Yelp-Airport | $10^{-3}$ | 128 |
| Yelp-Miss. | $10^{-2}$ | 64 |

Table 14: The best hyperparameter settings found for NHP on the forecasting task.

|  | Learning Rate | Embedding Dimension |
|---|---|---|
| PUBG | $10^{-3}$ | 128 |
| Reddit-C | $10^{-2}$ | 64 |
| Reddit-S | $10^{-2}$ | 64 |
| Taxi | $10^{-2}$ | 128 |
| Twitter | $10^{-2}$ | 128 |
| Yelp-Airport | $10^{-3}$ | 64 |
| Yelp-Miss. | $10^{-2}$ | 64 |

Table 15: The best hyperparameter settings found for diffusion on the unconditional generation task.

|  | Learning Rate | Weight Decay | Embedding Dimension | Layers |
|---|---|---|---|---|
| Hawkes1 | $10^{-3}$ | $10^{-6}$ | 64 | 2 |
| Hawkes2 | $10^{-2}$ | $10^{-5}$ | 64 | 4 |
| Nonstationary Poisson | $10^{-3}$ | $10^{-5}$ | 128 | 2 |
| Nonstationary Renewal | $10^{-3}$ | $10^{-4}$ | 64 | 2 |
| Stationary Renewal | $10^{-2}$ | 0 | 32 | 6 |
| Self-Correcting | $10^{-3}$ | 0 | 32 | 6 |
| PUBG | $10^{-3}$ | 0 | 64 | 2 |
| Reddit-C | $10^{-3}$ | $10^{-6}$ | 128 | 4 |
| Reddit-S | $10^{-3}$ | 0 | 64 | 4 |
| Taxi | $10^{-2}$ | 0 | 128 | 4 |
| Twitter | $10^{-3}$ | $10^{-4}$ | 64 | 6 |
| Yelp-Airport | $10^{-2}$ | 0 | 32 | 2 |
| Yelp-Miss. | $10^{-2}$ | $10^{-5}$ | 128 | 2 |

Table 16: The best hyperparameter settings found for diffusion on the forecasting task.

| | Learning Rate | Weight Decay | Embedding Dimension | Layers |
|---|---|---|---|---|
| PUBG | $10^{-4}$ | $10^{-5}$ | 32 | 6 |
| Reddit-C | $10^{-2}$ | $10^{-6}$ | 64 | 6 |
| Reddit-S | $10^{-3}$ | 0 | 64 | 4 |
| Taxi | $10^{-3}$ | $10^{-6}$ | 32 | 2 |
| Twitter | $10^{-4}$ | $10^{-5}$ | 64 | 6 |
| Yelp-Airport | $10^{-4}$ | $10^{-5}$ | 64 | 6 |
| Yelp-Miss. | $10^{-3}$ | $10^{-5}$ | 32 | 4 |

Table 17: The best hyperparameter settings found for Add-and-Thin on the unconditional generation task.

| | Learning Rate | Mixture Components |
|---|---|---|
| Hawkes1 | $10^{-3}$ | 32 |
| Hawkes2 | $10^{-2}$ | 32 |
| Nonstationary Poisson | $10^{-2}$ | 16 |
| Nonstationary Renewal | $10^{-2}$ | 8 |
| Stationary Renewal | $10^{-2}$ | 8 |
| Self-Correcting | $10^{-4}$ | 8 |
| PUBG | $10^{-3}$ | 8 |
| Reddit-C | $10^{-2}$ | 32 |
| Reddit-S | $10^{-2}$ | 16 |
| Taxi | $10^{-2}$ | 8 |
| Twitter | $10^{-4}$ | 32 |
| Yelp-Airport | $10^{-4}$ | 8 |
| Yelp-Miss. | $10^{-2}$ | 64 |

Table 18: The best hyperparameter settings found for Add-and-Thin on the forecasting task.

| | Learning Rate | Mixture Components |
|---|---|---|
| PUBG | $10^{-2}$ | 64 |
| Reddit-C | $10^{-2}$ | 16 |
| Reddit-S | $10^{-2}$ | 64 |
| Taxi | $10^{-2}$ | 8 |
| Twitter | $10^{-3}$ | 8 |
| Yelp-Airport | $10^{-2}$ | 32 |
| Yelp-Miss. | $10^{-3}$ | 16 |

