# OpenReview forum: "EventFlow: Forecasting Continuous-Time Event Data with Flow Matching"
_ICLR.cc/2025/Conference — Submitted to ICLR 2025_

### Official Review · Reviewer_u5Wj · 2024-10-28

**Soundness:** 3
**Presentation:** 3
**Contribution:** 2
**Rating:** 6
**Confidence:** 2

**Summary:**

The authors created a non-autoregressive model framework for continuous-time event sequences by using flow-matching. This allows them to skip the autoregressive process and also account for irregular time intervals. They achieve good results on a synthetic dataset and real world datasets compared to other baselines.

**Strengths:**

Paper is clear and is quite original with the framework of flow-matching; math is clear and has good structure.

**Weaknesses:**

I would've liked to see some more comparison against non-autoregressive type models for generation, such that it clarifies why the authors choose to use flow matching in particular. Comparatively, MMD wise the metric always measures with respect to the distribution, so with respect to this metric joint distribution models instead of autoregressive ones would have the advantage here.
I would like to see some more discussion on the synthetic datasets with IFTPP; why are the performances on the synthetic plot better as compared to the real world data?

**Questions:**

How does the length affect the corresponding accuracy? Is there an error vs sequence position plot?
Just a note that the appendix should be in the supplementary material and not the main portion :)

---

> ### Author Response · Authors · 2024-11-26
>
> We first would like to thank the reviewer for their valuable feedback, which we have used to improve the latest version of our paper.
>
> > I would've liked to see some more comparison against non-autoregressive type models for generation...
>
> We agree that additional non-autoregressive models certainly would be a useful point of comparison. However, to the best of our knowledge, our method EventFlow and Add-and-Thin (which we compare against) are the only two existing non-autoregressive models for unmarked TPPs. Designing and evaluating other classes of models within our framework of predicting the number of events $p(n | H)$ and their times $p_n(t_1, \dots, t_n)$ is an exciting direction for future work.
>
> That being said, we deliberately chose to build on the flow matching framework to design our model. In particular, flow matching has been shown to obtain strong results on complex, high-dimensional domains, e.g. images, while being simpler in terms of the necessary theory and implementation. Thus, our motivation was to obtain a simple yet highly performant model.
>
> > Comparatively, MMD wise the metric always measures with respect to the distribution, so with respect to this metric joint distribution models instead of autoregressive ones would have the advantage here.
>
> We would like to emphasize that the MMD metric is only used to evaluate our models when used for unconditional generation, and not forecasting. Moreover, on the forecasting task, where we do not use MMDs, we see a clear gap between the performance of autoregressive and non-autoregressive models.
>
> We also note that our primary focus is on multi-step forecasting tasks, and any evaluation of performance on this task naturally will involve the joint distribution over event times. One-step metrics, like MSE, are unable to capture the multi-step performance we are most interested in.
>
> > I would like to see some more discussion on the synthetic datasets with IFTPP...
>
> We thank the reviewer for noting that IFTPP performs better on the unconditional, synthetic task (Table 2). However, we would like to again highlight that our method performs best or second best on 5/6 of these datasets and obtains the highest overall mean rank for the unconditional generation task.
>
> As for why this is the case, our intuition is that this comes down to the complexity of the data -- the synthetic datasets are easier to model than the real-world data. On easier datasets, relatively simple models are sufficient for obtaining strong performance (e.g., IFTPP parametrizes the decoder as a log-normal mixture). The simplicity of the model in this case can be a boon, as simpler models should be more robust and easier to optimize. Conversely, on more difficult datasets, a more complex decoder becomes necessary, and the simplicity of IFTPP becomes a burden.
>
> > How does the length affect the corresponding accuracy?  Is there an error vs sequence position plot?
>
> We appreciate these interesting suggestions for additional ways of evaluating our method. We have performed an additional experiment, where we now evaluate the forecasting performance of our model across different forecasting horizons. In our preliminary results, we find that our method exhibits an approximately linear degradation in performance as we increase the value of $\Delta T$. In the table shared here, we report the forecasting errors (i.e., as in Table 1), where the columns indicate different values of $\Delta T$.
>
> We plan to perform the same experiment for the baselines to compare the rate at which the error increases, but have not had the time to complete this yet.
>
> | Dataset | 4 | 6 | 8 | 10 | 12 | 15 | 20 |
> |---------|---|---|---|-----|-----|-----|------------|
> | Reddit-C | 21.1 ± 1.1 | 28.6 ± 1.4 | 34.5 ± 2.1 | 39.3 ± 2.5 | 42.3 ± 2.3 | 46.6 ± 2.4 | 53.9 ± 2.4 |
> | Reddit-S | 28.0 ± 3.9 | 43.8 ± 6.5 | 65.6 ± 10.7 | 89.9 ± 14.3 | 112.8 ± 20.5 | 134.32 ± 23.9 | 138.6 ± 25.8 |
> | Taxi | 3.6 ± 0.1 | 5.1 ± 0.2 | 6.2 ± 0.4 | 7.6 ± 0.6 | 9.0 ± 0.9 | 9.6 ± 0.9 | 9.8 ± 0.7 |
> | Twitter | 1.7 ± 0.07 | 2.6 ± 0.1 | 3.3 ± 0.1 | 4.0 ± 0.2 | 4.5 ± 0.2 | 5.1 ± 0.2 | 5.5 ± 0.3 |
> | Yelp-A | 1.4 ± 0.02 | 1.7 ± 0.04 | 2.2 ± 0.1 | 2.7 ± 0.1 | 3.2 ± 0.2 | 3.6 ± 0.2 | 3.8 ± 0.1 |
> | Yelp-M | 2.0 ± 0.04 | 3.0 ± 0.04 | 4.0 ± 0.1 | 5.1 ± 0.1 | 6.1 ± 0.1 | 7.3 ± 0.1 | 7.5 ± 0.1 |
>
> | Dataset | 5 | 8 | 11 | 14 | 17 | 21 | 25 |
> |---------|---|---|---|-----|-----|-----|------------|
> | Pubg | 2.2 ± 0.02 | 2.9 ± 0.04 | 3.4 ± 0.1 | 3.8 ± 0.1 | 4.0 ± 0.1 | 4.2 ± 0.1 | 4.2 ± 0.1 |
>
>
> While we were unable to investigate how sequence position effects error for this response (due to the limited time allotted), we hope to investigate this further for a camera-ready version of our paper if accepted.
>
> > appendix should be in the supplementary material
>
> Please see the FAQ here: https://iclr.cc/Conferences/2025/AuthorGuide -- we opt to have all of our materials in one .pdf file for ease of access.

---

> > ### Comment · Reviewer_u5Wj · 2024-12-01
> >
> > Thanks for the comprehensive rebuttal and experiments.
> >
> > I'd like to just reassure the authors that I have read their rebuttal and have no further questions on my end, but since I'm more unfamiliar with this topic I'd like to read the other discussions before making a final decision on my score recommendation.

---

### Official Review · Reviewer_u733 · 2024-10-31

**Soundness:** 4
**Presentation:** 3
**Contribution:** 2
**Rating:** 6
**Confidence:** 4

**Summary:**

The paper proposes EventFlow, a flow-matching model for temporal point processes. EventFlow is evaluated on conditional and unconditional generative tasks. In contrast to previous diffusion-based approaches, balanced couplings are used, and the number of events does not change during generation. Therefore, the inference process consists of multiple steps: (1) EventFlow samples (unconditional) or predicts (conditional) the number of events. (2) A prior sequence $\gamma_0$ is sampled from a mixed-binomial TPP. (3) The sequence $\gamma_1$ is computed by solving the ODE with the vector field $v_\theta$.

In an experimental evaluation, EventFlow shows promising results and consistently outperforms multiple baselines.

**Strengths:**

- EventFlow models the event times of sequences via a CFM. The resulting model is more straightforward, more elegant, and effective than previous diffusion-based approaches.
- The evaluation includes a reasonable number of experiments, including unconditional and conditional comparisons on multiple datasets.
- The results show strong improvements compared to diffusion-based baselines.
- The paper is easy to follow, and the methodology is clearly described.
- The baselines are reasonably tuned.

**Weaknesses:**

- The biggest weakness, in my opinion, results from the use of balanced couplings. Modeling the event count distribution $p_\theta(n\mid \mathcal{H})$ with a deterministic classifier limits the distributional forecast, which is not the case for AddAndThin. I assume this would be more noticeable in a likelihood-based evaluation.
- The paper does not discuss whether a likelihood evaluation is possible. As EventFlow is based on a CNF, evaluating the likelihoods of single sequences should be possible. Including an NLL-based comparison, as done in other works (Shchur et al., 2019), would strengthen the evaluation.
- Certain hyperparameters are missing. How many neural function evaluations are used to solve the ODE? An ablation would help compare the performance with previous diffusion-based approaches.
- A runtime comparison with AddAndThin needs to be included. Especially as EventFlow uses an attention-based model.

Minors:
- page 4 footnote: i.e. -> i.e.,
- Shchur et al. (2019) is cited twice (arxiv and ICLR)

**Questions:**

- Is computing the likelihood for single sequences possible?
- Are the prior sequences ordered? I.e., can the paths of events cross when interpolating between $\gamma_0$ and $\gamma_1$?
- Are the quantitative results of the classifier $p_\theta(n\mid \mathcal{H})$?

---

> ### Author Response · Authors · 2024-11-26
>
> We thank the reviewer for their feedback, which we have incorporated into the latest version of our paper.
>
> > The biggest weakness, in my opinion, results from the use of balanced couplings. Modeling the event count distribution  pθ(n∣H) with a deterministic classifier limits the distributional forecast...
>
> We believe there might be a slight misunderstanding here. While the event count distribution $p_\phi(n | \mathcal{H})$ is learned as a classification problem (i.e., by minimizing the cross-entropy loss), at generation time we do not use a deterministic prediction. That is, rather than taking the argmax of $p_\phi(n | \mathcal{H})$, we sample from the corresponding distribution over n. See, for instance, Line 5 in Algorithm 2, and the “Sampling” paragraph in Section 4.4
>
> > The paper does not discuss whether a likelihood evaluation is possible...
> > Is computing the likelihood for single sequences possible?
>
> In principle, we believe that one could evaluate likelihoods with our method via the change-of-variables formula. This requires evaluating the likelihood of a prior sequence, which in our setup, is an ordered sequence of i.i.d. random variables. The corresponding likelihood can be found through the corresponding order statistics. However, in our work to date we chose not to use a likelihood-based evaluation, as two of our four baselines (Add-and-Thin and diffusion) are unable to calculate likelihoods. We agree, though, that this could still be a useful evaluation metric. We plan to add likelihood results in an updated version of our paper, but this will require us to add additional baselines which can compute likelihoods, which we have not had time to implement and tune yet given for this response.
>
> > Certain hyperparameters are missing. How many neural function evaluations are used to solve the ODE? An ablation would help compare the performance with previous diffusion-based approaches.
>
> We apologize for this oversight in our submission. Our flow time $s$ takes values $s \in [0, 1]$. At training time, $s$ is not discretized, but rather sampled uniformly on $[0, 1]$. This allows us to use an arbitrary choice of ODE solver at sampling time. In practice, we use a Euler solver with 25 uniformly sampled time steps (i.e., 25 NFEs) when sampling. We note that our framework is agnostic to this choice, and other ODE solvers are readily applicable.
>
> We are also grateful for the suggestion to perform an ablation regarding the NFEs. Please also see our global response, we perform an additional ablation where we vary the number of discretization steps in the solver.
>
> > A runtime comparison with AddAndThin needs to be included...
>
> We agree that an efficiency comparison would be useful. However, we generally prefer not to report wall-clock times for several reasons. Such times are generally hardware-dependent, can vary based on server load, are challenging to standardize across different codebases, and often depend on implementation details which are independent of the core efficiency of the method. That being said, we generally find that our method has favorable runtimes compared to add-and-thin. For example, to generate 100 batches of 256 sequences on the Yelp-M dataset, add-and-thin required 741 seconds, whereas our method required only 125 seconds. We emphasize, though, that sampling speed efficiency was likely not prioritized in these implementations.
>
> We instead prefer to report the number of function evaluations (NFEs) required to generate a sample. This metric is independent of implementation details and specific architecture choices, and instead highlights the underlying efficiency of the core method.
>
> In particular, our method is able to obtain strong forecasting results using only a single NFE to sample a sequence given $n$. We emphasize that the reference implementation of Add-and-Thin uses 100 NFEs to generate a sequence, diffusion uses 1000 NFEs per generated event, and the autoregressive baselines (IFTPP, NHP) require one NFE per generated event. Please see our global response for additional details.

---

> > ### Author Response · Authors · 2024-11-26
> >
> > > Are the prior sequences ordered?
> >
> > Yes, your understanding is correct -- the prior sequences $\gamma_0$ are ordered. Sequences $\gamma_1$ coming from the training distribution are also ordered, and hence the interpolation in Equation (4) will also be ordered.
> >
> > While a small number of generated events are not ordered, this typically only happens when the generated events are close together in time. Since our main tool for evaluation is the distance in Equation (10), which is in spirit an MAE, this occasional mis-ordering does not have a significant effect on our results. To sanity check this, we re-ran our evaluations after ordering the outputs of our model, and found this only changed our results in the second decimal place. To put this into context, this change is smaller than the precision we report in the paper, and significantly smaller than the standard deviation in performance across random seeds.
> >
> > > Are the quantitative results of the classifier...
> >
> >  We agree that further evaluation of our learned event count distribution would be useful. Please see our global response for an evaluation in terms of MARE.

---

> > > ### Comment · Reviewer_u733 · 2024-11-27
> > >
> > > I thank the authors for their response and additional results. I think the ablation across varying numbers of NFEs improves the empirical evaluation.
> > >
> > > I have some follow-up questions.
> > >
> > > > ... we sample from the corresponding distribution over $n$ ...
> > >
> > > By distribution over $n$, do you mean the output of the MLP after a softmax operation?
> > >
> > > > This requires evaluating the likelihood of a prior sequence, which in our setup, is an ordered sequence of i.i.d. random variables. The corresponding likelihood can be found through the corresponding order statistics.
> > >
> > > Let me rephrase my question. Can you evaluate the likelihood of a prior sequence? Can you elaborate more on "The corresponding likelihood can be found through the corresponding order statistics." I don't expect the authors to include a likelihood-based evaluation within the rebuttal. However, discussing the possibility of likelihood evaluation would strengthen the methodology as it is a key limitation of AddAndThin.
> > >
> > > > the prior sequences $\gamma_0$ are ordered
> > >
> > > This should not be necessary, right? Wouldn't an unordered prior sequence simplify the likelihood evaluation?

---

> > > > ### Author Response · Authors · 2024-11-30
> > > >
> > > > > By distribution over $n$ do you mean the output of the MLP after a softmax operation?
> > > >
> > > > Yes, your understanding is correct. Given a history $\mathcal{H}$, we pass this history into our network for predicting $n$. The final layer of our network produces a logit which we softmax to produce the distribution $p_\phi(n \mid \mathcal{H})$. Given this conditional distribution over $n$, we sample a value $n \sim p_\phi(n \mid \mathcal{H})$ at generation time.
> > > >
> > > > > Can you evaluate the likelihood of a prior sequence? Can you elaborate more on ... order statistics.
> > > >
> > > > Yes -- we apologize for the lack of clarity in our previous response. The details depend on what exactly is meant by "the" likelihood. For simplicity let us consider the unconditional setting, i.e., generating a sequence without a given history. The conditional case is analogous, but requires conditioning on $\mathcal{H}$ throughout.
> > > >
> > > > To sample a prior sequence $\gamma_0 \sim \mu_0$, we sample $n \sim \mu_1(n)$ from the *true, empirical* count distribution, followed by sampling $n$ i.i.d. event times $t_0^1, t_0^2, \dots, t_0^n \sim q$ from a prior $q$ and sorting, so that $t_0^1 < t_0^2 < \dots < t_0^n$. The corresponding likelihood of any sequence under this prior, **conditioned on exactly $n$ events occurring**, is given by  the joint density of the corresponding order statistics, i.e., $$q(t_0^1, \dots, t_0^k \mid n) = n! \prod_{k=1}^n q(t_0^k) \delta[t_0^1 < t_0^2 < \dots < t_0^n].$$
> > > >
> > > > This is readily computed as $q$ is known. Given a data sequence $\gamma_1 = (t_1^1, \dots, t_1^n)$, our model's likelihood $p(t_1^1, \dots, t_1^n \mid n)$ can be computed using standard continuous normalizing flow techniques -- that is, by solving the continuous-time change of variables ODE
> > > >
> > > > $$\partial_s \log p_s(t_s^1, \dots, t_s^n \mid n) = - \text{tr}(\partial_{\gamma_s} v_\theta(\gamma_s, s))$$
> > > >
> > > > backwards in time for $s \in [0, 1]$. See, for instance, Equation 4 in the [FFJORD](https://arxiv.org/abs/1810.01367) paper, and note that this requires computing the prior likelihood. Intensity-based methods are able to compute this model likelihood via Equation 2 in our paper.
> > > >
> > > > Note that our method allows us to compute the related quantity $p(t_0^1, \dots, t_0^n, n) = \mu_1(n) p(t_1^1, \dots, t_1^n)$ if $\mu_1(n)$ is estimated empirically from the training data. While autoregressive models are able to compute $p(t_1^1, \dots, t_1^n \mid n)$  (i.e., Equation 2), computing the full, joint likelihood $p(t_0^1, \dots, t_0^n, n)$ for these models would require sampling as the event count distribution is only implicitly defined.
> > > >
> > > > >  discussing the possibility of likelihood evaluation would strengthen the methodology
> > > >
> > > > We agree, and we thank the reviewer for bringing up likelihoods in their review. We are a bit short on space in the main paper, but we are working on condensing the writing so that we can discuss these aspects.
> > > >
> > > > > the prior sequences $\gamma_0$ are ordered
> > > >
> > > > > This should not be necessary, right? Wouldn't an unordered prior sequence simplify the likelihood evaluation
> > > >
> > > > You are correct that ordering the sequences is not strictly necessary. Generally, both ordered and unordered distributions are valid notions for specifying TPPs -- see, e.g., Chapter 5 of Daley-Vere-Jones. However, evaluating (prior) likelihoods is straightforward even with ordered sequences (see our previous discussion).
> > > >
> > > > Note, moreover, that without ordering, the path-lengths of the interpolants in Equation 3 will become *longer*. Working with ordered sequences thus serves to shorten and simplify our interpolants. We hypothesize that using unordered sequences may result in worse forecasting performance, especially for small NFEs. We plan to perform an additional ablation using an unordered prior to verify this.

---

> > > > > ### Comment · Reviewer_u733 · 2024-12-03
> > > > >
> > > > > I thank the authors for their response. My concerns have mostly been addressed.

---

### Official Review · Reviewer_9ccG · 2024-10-31

**Soundness:** 3
**Presentation:** 3
**Contribution:** 4
**Rating:** 6
**Confidence:** 3

**Summary:**

This paper introduces EventFlow, a novel generative model for temporal point processes (TPPs). Unlike existing autoregressive models that predict events one at a time and suffer from compounding errors, EventFlow directly learns joint distributions over event times, enabling more accurate multi-step forecasting. The model is likelihood-free, making it easier to implement and sample from than existing methods. EventFlow's performance is evaluated on various synthetic and real-world datasets, consistently outperforming existing methods in both unconditional generation and multi-step forecasting tasks (forecasting in a given window) .

**Strengths:**

Originality: this is a novel generative model for temporal point processes that bypass the autoregressive paradigm; potential the second model in this category ( the first being diffusion point process by ludke et al.)

Quality: the paper is very technical and technicality comes from defining appropriate probability measures; the authors reaches to an important conclusion – proposition 1, based on which they proposed eventflow. The proposed method eventflow including interpolant construction and training, and sampling seems very solid (although I did check the full details.) The paper also examines the two important cases conditional forecasting and unconditional with respect to history and conducts experiments which demonstrate the effectiveness.

Clarity: it is well presented for the most part.

Significance: I think this is an interest line of research as it deviates from traditional autoregressive models for TPP and I believe it is worthy of investigation by our community.

**Weaknesses:**

My main concern/weakness is evaluation. Especially for forecasting tasks where a separate model is learned for the event count distribution pφ(n | H). The authors treat the problem as classification. I assume learning is from the training data, where in some target window we have {1,…,N} events as target. The learned pφ(n | H) is then used to sample a count n for a specific instance for further inference task. For the experiments, while the authors use MMD as a metric for sequence distance, I don’t know how the forecasted number of events compared with the ground truth. Similarly I am not entirely sure how the forecasting times compared with the ground truth. Some visual aids will be very helpful.

**Questions:**

Can the authors clarify at what flow time do the authors use as results reported for your experiments, since s is discretized.

---

> ### Author Response · Authors · 2024-11-26
>
> We thank the reviewer for their detailed review and valuable feedback.
>
> > My main concern/weakness is evaluation...
> > I don’t know how the forecasted number of events compared with the ground truth...
>
> We thank the reviewer for suggesting further evaluation of the learned event count distributions.
>
> In the unconditional experiments, we simply sample the number of events $n$ from the training distribution. There is no learning of the event count distribution required here, as this is in a sense the optimal thing to do.
>
> In the conditional experiments, we now need to forecast the number of events conditioned on the history. This step requires learning, where we adopted a simple approach where we learn $p_\phi(n \mid H)$ by minimizing a cross-entropy loss. We agree that further evaluation of our learned event count distribution would be useful. Please see our global response for an evaluation in terms of MARE.
>
>
> > Can the authors clarify at what flow time do the authors use as results reported for your experiments, since s is discretized.
>
> We apologize for this oversight in our submission. Our flow time $s$ takes values $s \in [0, 1]$. At training time, $s$ is not discretized, but rather sampled uniformly on $[0, 1]$. This allows us to use an arbitrary choice of ODE solver at sampling time. In practice, we use a Euler solver with 25 uniformly sampled time steps  (i.e., 25 NFEs) when sampling. We note that our framework is agnostic to this choice, and other ODE solvers are readily applicable to a given, trained model.
>
> Please also see our global response, we perform an additional ablation where we vary the number of discretization steps in the solver.

---

### Official Review · Reviewer_WUdp · 2024-11-03

**Soundness:** 3
**Presentation:** 2
**Contribution:** 3
**Rating:** 5
**Confidence:** 3

**Summary:**

The work proposes a generative approach of modelling (unmarked) temporal point process based on diffusion models. The work decompose the distributions of temporal pointe process into the joint distribution of the number of events in a TPP realization and the joint distribution of event time conditioned on the number of events via a diffusion model in the rectified flow fashion. The proposed approach can be conditioned on history information for generate future events. Evaluation results on both unconditional generation results on commonly used TPP datasets and conditional future events forecasting shows superior performance of the model over baselines.

**Strengths:**

The work has the following strengths:
* This work extends diffusion models to the domain of temporal point process with a clear motivation of modelling joint distributions of events. This motivation positions the work as a novel approach toward the challenging task of future events forecasting in a fixed time horizon instead of single next event prediction. Experiment results also show clear advantages of the proposed work in multi-events forecasting in a fixed time window over existing works.
* The decomposition of TPP distribution into the distribution of the number of events and the joint distributions of event times is also a novel approach toward TPP modelling.

**Weaknesses:**

The work has the following weakness:
* The presentation of the work has room of improvements:
    * It is not clear what’s the purpose of introducing balanced coupling of TPPs in Section 4.1. The interpolation between two event sequences can be well defined without introducing balanced coupling and the number of events is modelled separately. In other words, the presentations of the methods does not necessarily rely on the introduction of balanced coupling and the theoretical results of the iff condition for the balanced coupling set to be non-empty. It would be helpful to add some intuitive motivation for this section in the work.
    * In the current presentation of approach, the original contribution (Section 4.1, 4.2, separate modelling of event count and event time) and existing work (Sections 4.3, 4.4, mostly rectified flows[1]) are intertwined in one section. I would suggest the authors to consider either clearly separating their original contributions from the methods in existing works or more clearly stating their contributions in terms of methodology instead of using generic statements of contributions like a generative model for TPP.
* The work constrains the type of TPP it models to unmarked temporal point process without event category labels. The practical values of the work is limited.
* It is arguable that MSE/RMSE for next event prediction is one of the most important evaluation results for TPP models due to its wide existence in many existing TPP works [2, 3, 4], practical applicability, and easiness to compare between models. As the work is capable of conditional generation, not including this evaluation task results is disappointing.

References

[1] Liu, Xingchao, Chengyue Gong, and Qiang Liu. "Flow straight and fast: Learning to generate and transfer data with rectified flow." arXiv preprint arXiv:2209.03003 (2022).

[2] Yang, Chenghao, Hongyuan Mei, and Jason Eisner. "Transformer embeddings of irregularly spaced events and their participants." arXiv preprint arXiv:2201.00044 (2021).

[3] Shchur, Oleksandr, Marin Biloš, and Stephan Günnemann. "Intensity-free learning of temporal point processes." arXiv preprint arXiv:1909.12127 (2019).

[4] Xue, Siqiao, et al. "Easytpp: Towards open benchmarking the temporal point processes." arXiv preprint arXiv:2307.08097 (2023).

**Questions:**

1. Do the models for conditional and unconditional generation share parameters?
2. The EventFlow model does not guarantee the generated sequences of event times preserves the original order of $\gamma_0$ but simply relies on learning such prior from training data. Does the work anpply any post-processing to generated results to deal with this potential problem? If not, do the authors encounter situations where the generated event sequence times are not in an increasing order?
3. The author claims the approach is likelihood-free. Is the re-ordering of sampled $\gamma_0$ part of the reasons that make the model incapable of evaluating likelihood of event sequences? Is it possible to define the distribution which $\gamma_0$ is sampled from in a fashion such that the $t_s$ are naturally in an ascending order?
4. Can the approach be extended to model marked TPP with event category labels in trivial ways like using the the hidden states of the denoising neural networks $\v_\theta$ to predict event categories?

---

> ### Author Response · Authors · 2024-11-26
>
> We thank the reviewer for their detailed comments and valuable suggestions for improving our submission, which we have used to update the latest version of our work.
>
> > It is not clear what’s the purpose of introducing balanced coupling of TPPs in Section 4.1...
>
> We agree that the role of the balanced couplings was not made sufficiently clear in our submission. We have added additional clarification in the latest version of our submission in Section 4.2 and Section 4.3
>
> While the interpolation between any two fixed, given sequences does not rely on this coupling (Equations 3-4), the corresponding marginal path of sequence distributions does crucially rely on choosing a coupling, as well as the corresponding marginal vector field (Equation 6). In other words, choosing a coupling is a necessary step in constructing our model.
>
> In Section 4.1, we single out a particular class of couplings -- the balanced couplings -- which allow us to construct a model that does not change the number of events. That is, if we had used an unbalanced coupling, our model would have to somehow incorporate a variable number of events. We explicitly seek to avoid this complexity. Proposition 1 tells us that the only way to obtain such a balanced coupling is if the reference measure $\mu_0$ has the same event count distribution as the data measure $\mu_1$.
>
> > I would suggest the authors to consider either clearly separating their original contributions from the methods in existing works or more clearly stating their contributions...
>
> We thank the reviewer for this suggestion regarding improving the presentation of our approach. We have added additional details in Sections 4.3 and 4.4 to better clarify the relationship with existing works.
>
> Note that, while the interpolant construction in Section 4.3 builds on standard flow matching ideas, this is only possible due to our use of a balanced coupling, which allows us to work with a fixed number of events when constructing our interpolants.
>
> > The work constrains the type of TPP it models to unmarked temporal point process without event category labels. The practical values of the work is limited.
> > Can the approach be extended to model marked TPP with event category labels in trivial ways...
>
> While our work indeed focuses on unmarked TPPs, we respectfully disagree that the practical value of our work is limited. We explicitly chose to focus on the unmarked setting for several reasons. First, there are many practical applications which do not involve marks, e.g. demand forecasting in online services [1] or modeling neural spike trains [2], as well as the datasets we consider in the submission.
>
> Second, as we discuss in Section 1, our main contribution regards modeling the event times themselves, for which we see significant gains in performance. While we believe that incorporating marks into our model is straightforward, this aspect would not serve as an effective way of evaluating our primary contribution. For instance, as the reviewer suggests, standard techniques would allow us to learn a mark decoder on top of our history encodings, in essence learning a conditional distribution over marks given the event times.
>
> > It is arguable that MSE/RMSE for next event prediction is one of the most important evaluation results for TPP models...
>
> We emphasize that our primary motivation is multi-step forecasting problems. Evaluating MSEs for next event prediction is somewhat at odds with this goal, since in the multi-step problem we must predict both the number and times of the events. We note that the cited papers which use MSEs to evaluate all focus on the single-step setting, and hence the appropriateness of MSE as a metric.
>
> However, we provide in Table 7 (Appendix C) an evaluation of one-step-ahead MSEs. We follow the setup of our forecasting experiment, except we now report the corresponding one-step-ahead MSEs. Generally, all of the methods show similar results on this metric, despite there being clear differences between methods on the multi-step task. We believe this serves to further highlight the necessity of moving beyond single-step prediction tasks.
>
> > Do the models for conditional and unconditional generation share parameters?
>
> In our experiments, the conditional and unconditional models do not share parameters. However, this is an interesting idea, and sharing parameters across these tasks is relatively straightforward if one so desires. One approach is to use a fixed (or learnable) embedding corresponding to “no history” and training our EventFlow model on a mixture of conditional and unconditional sequences.

---

> ### Author Response · Authors · 2024-11-26
>
> > The EventFlow model does not guarantee the generated sequences of event times preserves the original order...
>
> Your understanding is correct -- we do not explicitly enforce that the model produces sequences that are increasing order. At evaluation time, we did not post-process the generated sequences to re-order them.
>
> While a small number of generated events are indeed not ordered, this typically only happens when the generated events are close together in time. Since our main tool for evaluation is the distance in Equation (9), which is in spirit an MAE, this occasional mis-ordering does not have a significant effect on our results. To sanity check this, we re-ran our forecasting evaluation (i.e., corresponding to Table 1) after ordering the outputs of our model. This only changed our results in Table 1 in the second decimal place. To put this into context, this change is smaller than the precision we report in the paper, and significantly smaller than the standard deviation in performance across random seeds.
>
> > The author claims the approach is likelihood-free...
>
> We would like to clarify that our method is “likelihood-free” in the sense that training is not based on a maximum likelihood procedure.
>
> In principle, we believe that one could evaluate likelihoods with our method via the change-of-variables formula. This requires evaluating the likelihood of a prior sequence, which in our setup, is an ordered sequence of i.i.d. random variables. The corresponding likelihood can be found through the corresponding order statistics. However, in our work to date we chose not to use a likelihood-based evaluation, as two of our four baselines (Add-and-Thin and diffusion) are unable to calculate likelihoods. We agree, though, that this could still be a useful evaluation metric. We plan to add likelihood results in an updated version of our paper, but this will require us to add additional baselines which can compute likelihoods, which we have not had time to implement and tune yet given for this response.
>
> Please see our discussion with Review u733 for further details regarding likelihood calculations.
>
> [1] Intermittent Demand Forecasting with Deep Renewal Processes, Turkmen et al., 2019
> [2] Analysis of Neuronal Spike Trains, Deconstructed, Aljadeff et al., 2016

---

### Author Response · Authors · 2024-11-26
**Global Response to All Reviewers**

We thank the reviewers for their detailed, constructive reviews and valuable suggestions for improving our work. We have updated our submission to incorporate these suggestions, and changes to our manuscript are highlighted as blue text.

## Forecasting Results

While preparing our response, a careful review of our results resulted in us discovering a discrepancy in our experimental analysis which affects the forecasting results originally presented in Table 1. In particular, the sequence distance (Equation 9) depends on a choice of time $T$, corresponding to the maximum allowed event time. Previous work (Lüdke 2023) and our baselines implemented this using the value of $T$ corresponding to the maximum allowed event time across all sequences in a dataset. However, when forecasting, sequences are not generated until this time $T$, but rather only on some subwindow $[T_0, T_E]$ with $T_{E} = T_0 + \Delta T$ being some given ending time. Our original results in Table 1 for our method were computed using $T_E$, resulting in overly optimistic results.

We believe that the version of the distance metric using $T_E$ is more suitable for evaluating forecasts, as the version using $T$ implicitly depends on the location of the forecasting window $[T_S, T_E]$ within the support $[0, T]$. That is, if a given forecast is simply translated from the forecast window $[T_S, T_E]$ to a new window $[T_S’, T_E’]$, the distance using $T$ will change. However, the distance using $T_E$ (respectively $T_E’$) will remain invariant.

We have recomputed Table 1 using $T_E$ for all methods. Overall, our method still obtains the best mean performance on 5/7 of the datasets, and obtains the second best performance on the remaining 2/7 datasets (PUBG, Reddit-C) with competitive scores. While the numerical results of this experiment have changed, our key conclusions do not, as these results still indicate our model obtains state-of-the-art forecasting performance.

## Event Count Evaluation

Reviewers 9ccG and u733 suggest isolating and evaluating the performance of the learned event count distribution, i.e., $p_\phi(n \mid H)$. To that end, we include in Table 6 (Appendix C) a table of the mean absolute relative error (MARE) between the true and forecasted sequence lengths. This metric serves to evaluate only the predicted number of events in a forecast.

Here, we find that Add-and-Thin has strong performance (mean rank: 1.3), whereas our method (mean rank: 3), diffusion (mean rank: 3.1) perform comparably, while IFTPP (mean rank: 3.6) and NHP lag slightly behind (mean rank: 4)

While our method has room for improvement, we emphasize that even though our approach to learning $p(n \mid H)$ is quite simple it still achieves competitive results. Designing better techniques for predicting the event counts is an exciting direction for future work, and would likely lead to even stronger forecasting results. We further emphasize that our model shows strong forecasting performance (Table 1), which simultaneously measures both the event counts and their times, and this is the primary relevant metric for the problem we address in this paper.

We also perform an ablation study (see Table 1) where we set the number of events $n$ to be the true number of events in a given forecast. This serves to isolate only the event time prediction component of our model, and goes to show that future work improving the predictions for $n$ can lead to significant forecasting gains.

## NFE Ablation

Reviewer u733 mentions the number of function evaluations (NFEs) used by our model at generation time (i.e., in Equation 9). We originally used 25 NFEs. Here, we perform an additional ablation on the choice of NFEs. To do so, we took our models used in the forecasting experiment and reduced the NFEs to 10 or 1 at generation time. The resulting forecast errors are reported in Table 1. We find that 10 NFEs is sufficient to obtain results on-par with those we obtained previously with 25 NFEs. Interestingly, with only a single NFE, we observe only a small drop in forecasting performance, with our model still obtaining the best forecasting performance on 4/7 datasets and second-best performance on the remaining 3/7 datasets. We emphasize that Add-and-Thin uses 100 NFEs at generation time, and the diffusion model uses 1000 NFEs per generated event. The remaining baselines (NHP, IFTPP) require one NFE per generated event. Thus, our method is able to simultaneously obtain strong forecasting performance while only requiring a small number of NFEs.

The level of performance obtained by our model with a small number of NFEs is enabled by our carefully designed interpolant construction (Equation 3). As these interpolants are linear, the resulting paths are typically straight and easy to integrate (i.e., requiring only a small number of steps in the differential equation solver).

---

### Meta-Review · Area_Chair_cLGm · 2024-12-22

**Metareview:**

This paper introduces EventFlow, a non-autoregressive generative model for continuous-time event sequences based on flow matching. EventFlow models the joint distribution of event times, avoiding the cascading errors of autoregressive methods. It demonstrates strong performance on unconditional and conditional forecasting tasks across several datasets. The paper proposes a new application of flow matching to temporal point processes (TPPs).

**Additional Comments On Reviewer Discussion:**

The reviewers noted the novelty and technical soundness of the proposed approach, particularly its deviation from traditional autoregressive models. However, concerns were raised regarding evaluation, such as the handling of event counts, the absence of a likelihood-based assessment, and runtime comparisons. Other concerns included the practicality of balanced couplings and the applicability of the framework to broader TPP scenarios. The authors addressed some of these concerns in their response, but reviewers did not reach a consensus and concerns about the evaluation methodology and the generalizability persisted. There was also lack of a clear champion for the paper.

---

### Decision · Program_Chairs · 2025-01-22

Reject